# UPT++: Latent Point Set Neural Operators for Modeling System State Transitions

## Abstract

Particle methods comprise a wide spectrum of numerical algorithms, ranging from computational fluid dynamics governed by the Navier-Stokes equations to molecular dynamics governed by the many-body Schrödinger equation. At its core, these methods represent the continuum as a collection of discrete particles, on which the respective PDE is solved. We introduce UPT++, a latent point set neural operator for modeling the dynamics of such particle systems by mapping a particle set back to a continuous (latent) representation, instead of operating on the particles directly. We argue via what we call the *discretization paradox* that continuous modeling is advantageous even if the reference numerical discretization scheme comprises particles. Algorithmically, UPT++ extends Universal Physics Transformers – a framework for efficiently scaling neural operators – by novel importance-based encoding and decoding. Furthermore, our encoding and decoding enable outputs that remain consistent across varying input sampling resolutions, i.e., UPT++ is a neural operator. We discuss two types of UPT++ operators: (i) time-evolution operator for fluid dynamics, and (ii) sampling operator for molecular dynamics tasks. Experimentally, we demonstrate that our method reliably models complex physics phenomena of fluid dynamics and exhibits beneficial scaling properties, tested on simulations of up to 200k particles. Furthermore, we showcase on molecular dynamics simulations that UPT++ can effectively explore the metastable conformation states of unseen peptide molecules.

## 1 Introduction

In both science and engineering, substantial efforts have led to the formulation of complex mathematical models that accurately represent physical phenomena. Prominent examples include the Navier-Stokes equations, which describe fluid dynamics, and the Schrödinger equation, fundamental to quantum mechanics.

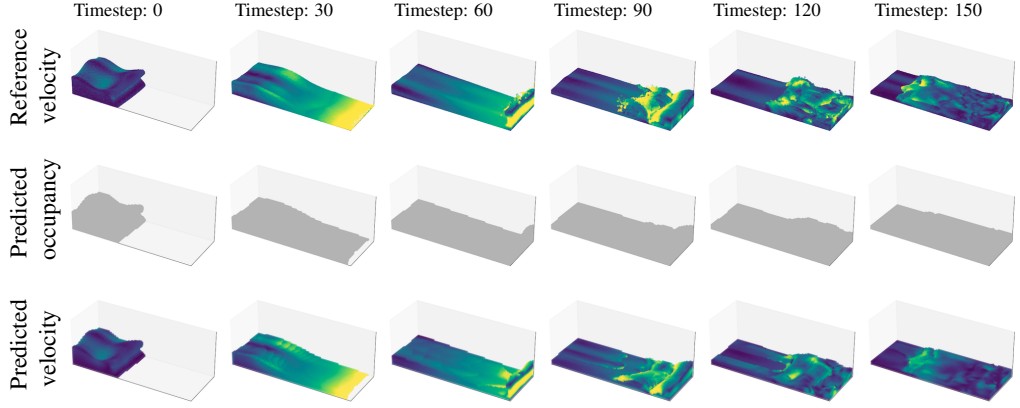

Figure 1: Visualization of a DamBreak3D trajectory. UPT++ successfully captures the characteristics of the evolving fluid, both in terms of predicted occupancy field, and – conditioned on the occupancy field – predicted velocity field. Lighter colors correspond to higher absolute velocities.

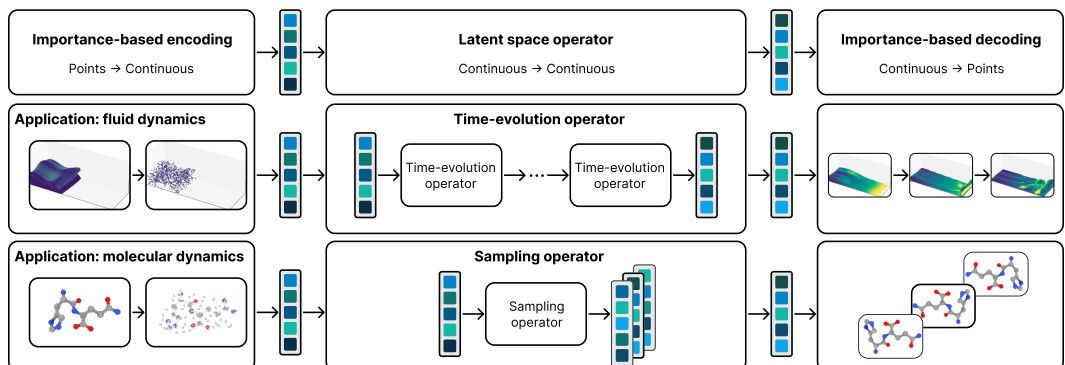

Figure 2: **The UPT++ modeling paradigm.** UPT++ encodes point-cloud information into a continuous latent space representation and decodes this representation at arbitrary query points. This framework enables new ways of simulating particle systems. For example, in our fluid dynamics experiments, particle velocities are sampled at particle positions, whereas in our molecular sampling experiments, densities around atoms are sampled and encoded. The respective latent space operators model the time evolution of the fluid or allow to sample new conformations, respectively. The resulting latent representations are point-wise decoded to occupancies and the corresponding physics information.

The Navier-Stokes equations are the cornerstone of fluid mechanics, and – despite being formulated in the 19th century – continue to present significant challenges to mathematicians and physicists. Most notably, the proof of existence and smoothness of solutions to the Navier-Stokes equations in three dimensions is one of the seven "Millennium Prize Problems" set by the Clay Mathematics Institute (Carlson et al., 2006), with a 1 million prize offered for a solution. What the Navier-Stokes equations are for fluid mechanics, the Schrödinger equation is for quantum mechanics, i.e., the fundamental building block to describe the behavior of particles at atomic and subatomic scales. Unlike classical mechanics, which deals with deterministic paths, the Schrödinger equation (Schrödinger, 1926) embraces the probabilistic nature of quantum phenomena, allowing for the superposition of states and the emergence of phenomena like quantum entanglement and tunneling.

Solving PDEs numerically is also not straightforward, particularly due to the potential for numerical instabilities, which can lead to inaccurate results or convergence issues. A successful class of numerical methods to solve certain types of PDEs are particle methods (Pahlke & Sbalzarini, 2023), which represent the underlying continuum media as a collection of discrete particles. For example, for many complex phenomena modeled by the Navier-Stokes equations, e.g., free surface dynamics, or multi-phase flows, Lagrangian discretization schemes are prevalent. Lagrangian methods employ finite material points, often termed particles, whose movement aligns with the local deformation of the continuum (Gingold & Monaghan, 1977; Lucy, 1977; Cundall & Strack, 1979; Brackbill & Ruppel, 1986). Similarly, for molecular dynamics, the Born-Oppenheimer approximation (Born & Oppenheimer, 1927) separates the dynamics of electrons and nuclei, which allows one to move the nuclei – when seen as point particles – according to the laws of classical Newtonian dynamics.

**The discretization paradox**. Contrasting numerical discretization schemes and recent advances in deep learning reveal a subtle paradox. While particle-based discretization schemes are often used to model the most complex natural phenomena, deep learning shines at learning continuous representation, e.g., neural fields (Sitzmann et al., 2020; Mildenhall et al., 2021; Xie et al., 2022), at modeling continuous transformation, e.g., diffusion models (Ho et al., 2020) and flow matching (Lipman et al., 2022), and at continuous modulation (Perez et al., 2018; Peebles & Xie, 2023). Naturally, the question arises, why – if the underlying nature is continuous anyway – we don't leverage the power of deep learning on continuous representations?

Towards this end, we introduce UPT++, which builds on Universal Physics Transformers (Alkin et al., 2024), a latent space neural operator framework for efficiently scaling neural operators to larger physics systems. UPT++ extends Universal Physics Transformers by importance-based encoding and importance-based decoding schemes, which allows us to convert point cloud representations of particle methods into continuous latent space representations. Most importantly, encod-

ing and decoding enable outputs that remain consistent across varying input sampling resolutions, which qualifies UPT++ as a neural operator. The UPT++ operator types we are discussing are time-evolution operators – as used for the time-evolution of the Navier-Stokes equations, and sampling operators – applicable to molecular dynamics simulations.

Our contributions are summarized as follows:

- We connect particle-based numerical discretization schemes and latent space modeling. UPT++ allows us to suggest new modeling paradigms – demonstrated on particle-based fluid dynamics and molecular dynamics simulations.
- We introduce novel importance-based encoding and importance-based decoding schemes to switch between discretized physics space and continuous latent space.
- We demonstrate the efficacy and scaling properties of UPT++ on fluid simulations of up to 200k particles, and strong sampling performance on molecular dynamics data.

## 2 BACKGROUND: PARTICLE METHODS AND NEURAL OPERATORS

**Example: SPH discretization in fluid dynamics**. The incompressible Navier-Stokes equations (Temam, 2001) are defined for the velocity flow field $\boldsymbol{u} : \mathcal{X} \times [0, T] \to \mathbb{R}^3$, $\mathcal{X} \subset \mathbb{R}^3$, and entail momentum and mass conservation, i.e., $\rho \frac{\mathrm{d}\boldsymbol{u}}{\mathrm{d}t} = \mu \nabla^2 \boldsymbol{u} - \nabla p + \rho \boldsymbol{f}$, and $\nabla \cdot \boldsymbol{u} = 0$, respectively. Here, $\rho$ is the density, $\mathrm{d}\boldsymbol{u}/\mathrm{d}t$ is the material derivative, i.e., the rate of change of $\boldsymbol{u}$ of a material element, $\mu \nabla^2 \boldsymbol{u}$ is the viscosity, i.e., the diffusion of $\boldsymbol{u}$ modulated by the viscosity parameter $\mu$, $p$ is the pressure, and $\boldsymbol{f}$ an external force. Lagrangian discretization schemes discretize the continuum via finite material points that move according to the local deformation of the continuum. A famous example is given by smoothed particle hydrodynamics (SPH) proposed by Lucy (1977) and Gingold & Monaghan (1977). SPH approximates the field properties using radial kernel interpolations over adjacent particles at the location of each particle. The strength of the SPH method is that it does not require connectivity constraints, e.g., meshes, which is particularly useful for simulating systems with large deformations, complex fluid-boundary interactions, or free surface flows.

**Example: Nuclei discretization in molecular dynamics**. The temporal evolution of quantum systems is governed by the many-body Schrödinger equation, for which analytic solutions are hardly known and are only available for simple systems like free particles or hydrogen atoms. Classical molecular dynamics (MD) enables the simulation of large atomic systems by various approximations reducing the degrees of freedom used to describe the system. A well-known example is the Born–Oppenheimer approximation, where the time-evolving solution is separated into components for the heavier atomic nuclei and the lighter, faster-moving electrons. Further, the positions of the nuclei are updated using numerical integration, where forces on the nuclei are obtained via the negative gradient of the potential energy given by an approximate solution of the electronic Schrödinger equation. The potential energy is a parameterized sum of intra- and intermolecular interaction terms; we refer to Appendix B.1 and Frenkel & Smit (2002) for more details on MD. A common application of MD is for sampling conformation states of biomolecules from the Boltzmann distribution, which is also an active area of research in machine learning (Noé et al., 2019). From ergodic theory, we know that, in most cases, MD generates samples from the Boltzmann distribution when the simulation is long enough (Frenkel & Smit, 2002), which, unfortunately, for biomolecules often means simulating for $10^{15}$ integration steps. Data-driven approaches could accelerate this sampling process by either simulating with larger integration steps or directly sampling from the target distribution.

**Operator learning**. The operator learning paradigm (Lu et al., 2019; 2021; Li et al., 2020b;a; Kovachki et al., 2021) targets the approximation of mappings between function spaces. Such function spaces can e.g., comprise the solutions of partial differential equations (PDEs). Following Kovachki et al. (2021), we assume $\mathcal{U}, \mathcal{V}$ to be Banach spaces of functions on compact domains $\mathcal{X} \subset \mathbb{R}^{d_x}$ or $\mathcal{Y} \subset \mathbb{R}^{d_y}$, mapping into $\mathbb{R}^{d_u}$ or $\mathbb{R}^{d_v}$, respectively. Operator learning aims to approximate the ground truth operator $G : \mathcal{U} \to \mathcal{V}$ via $\hat{\mathcal{G}} : \mathcal{U} \to \mathcal{V}$, typically framed as a supervised learning problem, where input-output pairs are i.i.d. sampled. However, the notable difference is that in operator learning, the space sampled from is not finite-dimensional. More precisely, with a given data set consisting of $N$ function pairs $(\boldsymbol{u}_i, \boldsymbol{v}_i) = (\boldsymbol{u}_i, \mathcal{G}(\boldsymbol{u}_i)) \subset \mathcal{U} \times \mathcal{V}$, $i = 1, ...N$, we aim to learn $\hat{\mathcal{G}} : \mathcal{U} \to \mathcal{V}$, so that $\mathcal{G}$ can be approximated in a suitably chosen norm.

A widely adopted approach is to approximate $\mathcal{G}$ via three maps (Seidman et al., 2022; Alkin et al., 2024): $\mathcal{G} \approx \hat{\mathcal{G}} := \mathcal{D} \circ \mathcal{A} \circ \mathcal{E}$, comprising encoder $\mathcal{E}$, approximator $\mathcal{A}$, and decoder $\mathcal{D}$. In recent works (Wang et al., 2024; Alkin et al., 2024), the decoder $\mathcal{D}$ has been formulated via point-wise queries at the output grid or mesh. In this work, we focus on two instances of the approximator $\mathcal{A}$:

- Time-evolution operators (Seidman et al., 2022; Alkin et al., 2024; Wang et al., 2024), which approximate the deterministic time evolution of a system.
- Sampling operators, which are trained to represent the molecular conformation space (Xu et al., 2022; Jing et al., 2022).

**Neural operators for particle systems**. Whereas most state-of-the-art neural operator methods are tailored for grid-based, predominantly regular domains, neural operator formulations for particle- or mesh-based dynamics remain limited. In such cases, graph neural networks (GNNs) (Scarselli et al., 2008; Kipf & Welling, 2017) with graph-based latent space representations offer a promising alternative. Often, predicted node accelerations are numerically integrated to simulate the time evolution of multi-particle systems (Sanchez-Gonzalez et al., 2020; Mayr et al., 2023; Toshev et al., 2023a). GNNs inherently possess a strong inductive bias for Lagrangian dynamics, which, however, also presents a significant downside since the number of nodes, and thus the computational complexity grows with the number of Lagrangian particles. Thus, computational complexity becomes quickly infeasible for an increasing number of particles (Alkin et al., 2024; Musaelian et al., 2023), see Figure 3. Furthermore, the effective degrees of freedom of a particle system are sometimes orders of magnitude less than the degrees of freedom arising from the discretization, especially in simulations showcasing bulk behavior. Lastly, for particle systems, a neural operator formulation is harder, especially with respect to the discretization convergence property (Li et al., 2020a), whereas a neural network is expected to show consistency for increasing input sampling resolutions.

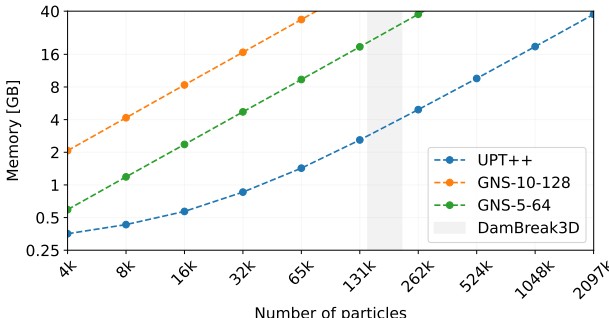

Figure 3: Qualitative exploration of scaling limits when modeling particle systems. Starting from 4k input points, we compared training time memory requirements of popular Graph Network-based Simulators (Sanchez-Gonzalez et al., 2020) against UPT++. We compare two GNS versions of 5 and 10 layers with hidden dimensions of 64 and 128, respectively. The tested UPT++ model has 30M parameters and can be trained with up to 2M particles on a single A100 40GB GPU.

## 3 UPT++

Our work builds on Universal Physics Transformers (UPTs) (Alkin et al., 2024) – a recently introduced framework for scaling neural operators, which follows the encoder-approximator-decoder approach. UPT flexibly encodes different grids, and/or a different number of particles into a unified latent space representation and introduces a decoder that queries the latent representation at different locations. We adopt the approach of composite neural operators $\mathcal{G} \approx \hat{\mathcal{G}} := \mathcal{D} \circ \mathcal{A} \circ \mathcal{E}$, where $\mathcal{E} : \mathcal{U} \rightarrow \mathbb{R}^{n_{\text{latent}} \times h}$ maps a solution state $\boldsymbol{u}^t \in \mathcal{U}$ to a latent space state representation $\boldsymbol{z}^t := \mathcal{E}(\boldsymbol{u}^t) \in \mathbb{R}^{n_{\text{latent}} \times h}$, i.e., to $n_{\text{latent}}$ tokens of dimension $h$, $\mathcal{A} : \mathbb{R}^{n_{\text{latent}} \times h} \rightarrow \mathbb{R}^{n_{\text{latent}} \times h}$ maps the latent state $\boldsymbol{z}^t$ to a successor latent state $\boldsymbol{z}^{t'} := \mathcal{A}(\boldsymbol{z}^t) \in \mathbb{R}^{n_{\text{latent}} \times h}$, $t' > t$, and $\mathcal{D} : \mathbb{R}^{n_{\text{latent}} \times h} \rightarrow \mathcal{U}$ reconstructs a solution state $u^{t'} \in \mathcal{U}$ from the latent space state representation $\boldsymbol{z}^{t'} \in \mathbb{R}^{n_{\text{latent}} \times h}$, i.e. $\boldsymbol{u}^{t'} := \mathcal{D}(\boldsymbol{z}^{t'}) \in \mathcal{U}$.

**Deterministic time-evolution: application to fluid dynamics**. We consider fluid dynamics problems where the fluid is contained within a domain $\Omega \subset \mathcal{X} \subset \mathbb{R}^3$ but does not fill the entire domain $\mathcal{X}$. The regions of the domain occupied by the fluid $\Omega$ change over time governed by the velocity field of the fluid at each given time. Equivalently, we can consider that we are given two disjoint fluid sets (e.g., air and water) filling the entire domain. Then, we consider the solution state $\boldsymbol{u}^t$ to be a compound function, consisting of a velocity field $\boldsymbol{v}^t$ and an occupancy field $\boldsymbol{o}^t$, i.e., $\boldsymbol{u}^t = (\boldsymbol{v}^t, \boldsymbol{o}^t)^T$. The velocity field $\boldsymbol{v}^t$ maps a certain coordinate $\boldsymbol{x} \in \mathcal{X}$ to a point-wise velocity at this coordinate, i.e., $\boldsymbol{v}^t : \mathcal{X} \mapsto \mathbb{R}^3$. The occupancy field $\boldsymbol{o}^t$ maps a certain coordinate $\boldsymbol{x} \in \mathcal{X}$ to whether a fluid particle is at this coordinate or not, i.e., $\boldsymbol{o}^t : \mathcal{X} \mapsto \{0, 1\}$. $\boldsymbol{u}^t$ is therefore a function $\boldsymbol{u}^t : \mathcal{X} \mapsto \mathbb{R}^3 \times \{0, 1\}$.

**Stochastic conformation sampling: application to molecular dynamics**. We assume that a molecule is spatially located in $\mathcal{X} \subset \mathbb{R}^3$ and the specific conformation of a certain molecule is represented by a vector of continuous density fields, i.e., each component of $\boldsymbol{u}^t$ represents one specific density field associated with the conformation of the molecule at time $t$. Each density field represents some specific characteristics of the molecular conformation, i.e., it might, for example, be specific to a certain atom type. We construct each density field analogously to Pinheiro et al. (2024) and Dumitrescu et al. (2024). Appendix B.2.1 gives further details on the construction of the density fields. When we make use of $d$ density fields to represent the overall conformation of a molecule, then $\boldsymbol{u}^t$ is a function $\boldsymbol{u}^t : \mathcal{X} \mapsto \mathbb{R}^d$.

When applying UPT to such settings, we need to address three main questions, leading to UPT++:

1. How to encode complex point-cloud representations into continuous representations, and how to decode point-wise from continuous representations?

2. How to formulate deterministic and sampling operators in the latent space?

3. How to efficiently train UPT++?

### 3.1 IMPORTANCE-BASED ENCODING, IMPORTANCE-BASED DECODING

**Importance-based encoding**. We introduce importance-based encoding, which consists of four conceptual steps, see Figure 4. We start with *occupancy-based selection*, which accounts for the fact that in many simulations, disjoint sets fill the entire domain. For example, for many fluid dynamics phenomena, the state $\boldsymbol{u}^t$ is a compound state of a materialized fluid field, and either obstacles or a second fluid, potentially in the gaseous phase. Similarly, for molecular dynamics, molecules are represented within a box, where not the entire box is filled. Secondly, via *importance-based sampling* we emphasize information within the occupied regions. For example, a sampling strategy for particle methods could be to upsample denser regions to account for the larger concentration of mass there. Similarly, we sample points around atoms according to density fields consisting of 3D Gaussian density distribution centered at each atomic position. After importance-based sampling, the encoder $\mathcal{E}$ first embeds the selected $k$ points into hidden dimension $h$, adding positional encoding (Vaswani et al., 2017) to the different nodes, i.e., $\boldsymbol{u}_k^t \in \mathbb{R}^{k \times d} \to \mathbb{R}^{k \times h}$. Next, via *local aggregation* we propagate neighboring information to the respective supernodes (radius graph for connectivity to keep discretization convergence), and finally *global information aggregation* pools the information into a fixed size and uniform latent space via perceiver blocks (Jaegle et al., 2021a;b). The resulting continuous latent space contains $n_{\text{latent}}$ latent tokens of dimension $h$, i.e., $\boldsymbol{z}^t := \mathcal{E}(\boldsymbol{u}^t) \in \mathbb{R}^{n_{\text{latent}} \times h}$.

**Importance-based decoding**. Importance-based decoding reverses the conceptual steps of importance-based encoding. First, via *occupancy decoding* (Mescheder et al., 2019), we decode an occupancy field to identify where particles or atoms are located in space. Secondly, we *point-wise decode* a field quantity and consider only the occupied points. For example, for fluid dynamics, we consider only the flow velocity in regions where occupancy is predicted. Similarly, for molecular dynamics, we decode whether regions are occupied, and consider our density predictions on those. Analogous to Alkin et al. (2024), the decoder is implemented via a perceiver-like cross-attention layer using a positional embedding of the output positions as query and the latent representation as keys and values. Since there is no interaction between queries, the latent representation can be queried at arbitrarily many positions without large computational overhead. To train the occupancy field and enable positive and negative learning signals, we sample points at all locations of the domain. During inference, we simultaneously predict the occupancy field and the corresponding field quantities and only keep the field at occupied points. For the reconstruction of molecular graph structures, we adopt a method similar to that of Pinheiro et al. (2024); Dumitrescu et al. (2024),

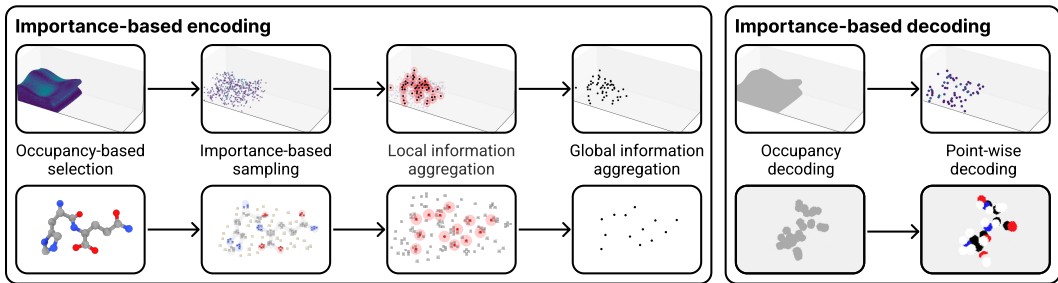

Figure 4: Importance-based encoding and decoding schemes of UPT++, which allow us to encode complex point-cloud representations into continuous representations, and reversely enable point-wise decoding of continuous representations.

utilizing an efficient peak-finding algorithm to identify atom positions, and OpenBabel (O'Boyle et al., 2011) to reconstruct the corresponding molecular bonds.

## 3.2 LATENT SPACE APPROXIMATOR

In UPT++, $\mathcal{A} : \mathbb{R}^h \to \mathbb{R}^h$ is a latent operator, which maps the latent state $z^t$ to a successor latent state $z^{t'} := \mathcal{A}(z^t) \in \mathbb{R}^h$ (in the case of a deterministic *time-evolution* operator) or which samples a successor latent state $z^{t'} \in \mathbb{R}^h$ from a conditional probability distribution $p(.|z^t)$, i.e., $z^{t'} \sim p(.|z^t)$ (in the case of a stochastic *sampling operator*).

**Time-evolution operator**. The time-evolution operator, implemented as a transformer (Alkin et al., 2024), $\mathcal{A} : z^t \in \mathbb{R}^{n_{\text{latent}} \times h} \to z^{t'} \in \mathbb{R}^{n_{\text{latent}} \times h}$, propagates the compressed representation forward in time. As $n_{\text{latent}}$ is small, forward propagation in time is fast. Notably, the approximator can be applied multiple times, propagating the signal forward in time by $\Delta t$ corresponding to each call of the approximator. After inferring one timestep, it is not necessary to decode the latent state and encode it again to compute the result of a further timestep. Instead, in inference mode, we can keep the latent representation and apply the forward operator again. We call this process *latent rollout*. Especially when working with many particles, the benefits of latent space rollouts, i.e., fast inference, pay off. However, to enable latent rollouts, the responsibilities of encoder $\mathcal{E}$, approximator $\mathcal{A}$, and decoder $\mathcal{D}$ need to be decoupled, which is further discussed in Section 3.3. It should be mentioned that our integration timestep $\Delta t$ is a multiple of the simulation steps used for dataset generation (sometimes up to a factor thousand larger).

**Sampling operator**. The sampling operator that is implemented via flow matching (Lipman et al., 2022), samples $z^{t'}$ from a conditional distribution $p(.|z^t)$, i.e., sampling is conditioned on the latent state $z^t$. The learning objective for the approximator is to construct a parameterized ($\theta$) distribution $p_\theta(.|z^t) \approx p(.|z^t)$. At inference, we draw samples from $p_\theta(.|z^t)$. Thereby we assume an atomistic timestep $\Delta t$, which is the minimum timestep for which $p_\theta(.|z^t)$ was trained to generate meaningful predictions. $\Delta t$ is usually equal to or a multiple of simulation timesteps. The actual time difference $t' - t$ between prediction time and condition time is usually again a multiple of $\Delta t$. We implement rectified flow match (Liu et al., 2022) with adaptions according to Li et al. (2024) to include the conditioning on $z^t$ using classifier-free guidance (Ho & Salimans, 2022). Appendix B.3 details the training and sampling procedures of the UPT++ sampling operator. In contrast to the time-evolution operator, we extend the sampling operator to also explicitly depend on the number of atomistic timesteps, i.e., we aim to directly learn $p_\theta(z^{t'}|z^t, N) = p_\theta(z^{N \Delta t + t}|z^t, N) \approx p(z^{t'}|z^t)$ to draw $z^{t'}$ after $N \Delta t$ steps instead of drawing a chain of $N$ consecutive samples $z^{t+\Delta t} \sim p(.|z^t), \ldots, z^{t+N \Delta t} \sim p(.|z^{t+(N-1) \Delta t})$.

## 3.3 UPT++ TRAINING PROCEDURE

**Training**. We make use of the decomposition $\mathcal{D} \circ \mathcal{A} \circ \mathcal{E}$ and split up training into 2 stages, keeping in mind the motivation to enable latent rollouts, for which the responsibilities of encoder $\mathcal{E}$, approximator $\mathcal{A}$, and decoder $\mathcal{D}$ need to be decoupled. Therefore, at the *first training stage*, we

Figure 5: Sketch of the two types of latent operators we use: PDE forward operator for modeling the time-evolution of the Navier-Stokes equations and the sampling operator applicable to molecular dynamics simulations.

train $\mathcal{E}$ and $\mathcal{D}$ by sampling $k$ input points, and – not necessarily related – $k'$ output points. At this stage, we don't apply the forward operator (therefore $t' = t$), and the encoder-decoder training can be considered as the training of an autoencoder for the compound state $\boldsymbol{u}^t$. In the *second training stage*, we freeze the encoder and the decoder weights, and make use of different timesteps $t$ and $t'$, $t' > t$, to only train the approximator $\mathcal{A}$ given the fixed latent space input $\boldsymbol{z}^t$ and target output $\boldsymbol{z}^{t'}$. Additionally, for training the latent sampling operator, we regularize the latent space via KL-divergence (Zhang et al., 2023; Rombach et al., 2022).

### 3.4 RELATED WORK

In recent years, deep learning has started to make a significant impact in the field of computational fluid dynamics (Guo et al., 2016; Li et al., 2020a; Thuerey et al., 2021; Kochkov et al., 2021; Vinuesa & Brunton, 2022; Gupta & Brandstetter, 2022; Lam et al., 2022; Bi et al., 2022; Brandstetter et al., 2022b;a; Andrychowicz et al., 2023; Bodnar et al., 2024; Herde et al., 2024). Several of those works have applied Transformers to physical systems. Galerkin Transformer (Cao, 2021) uses Galerkin-type attention to address attention complexity, GNOT (Hao et al., 2023) employs linear attention, OFormer (Li et al., 2023a) uses recurrent MLPs to propagate solutions over time and FactFormer (Li et al., 2023b) uses multidimensional factorized attention. However, all these methods apply attention directly to the input points, which is not scalable in our setting. Transolver (Wu et al., 2024) reduces the number of tokens by learning a mapping to physics-aware tokens, a concept similar to our use of supernodes. However, their mapping is recomputed in each Transformer layer, whereas we operate within a fixed latent space. OFormer (Li et al., 2023a) also employs a positional embedding combined with a perceiver to query at arbitrary points and perform decoding. However, their approach applies this process directly to the input, followed by a push forward operation, resulting in fixed queries that cannot be altered, thus not suitable for our problem setting. CViT (Wang et al., 2024) is the most similar to our decoding method, but it replaces positional embeddings with learned grid features and uses interpolation to generate the queries.

Recent advancements in deep learning have led to an increased interest in its application to molecules. *Boltzmann generators* (Noé et al., 2019; Köhler et al., 2021) employ flows to draw asymptotically unbiased samples from the Boltzmann distribution, but lack the ability to generalize across multiple molecules. Unlike UPT++, current approaches apply sampling either in torsion (Jing et al., 2022) or Euclidean space (Klein et al., 2024a;b; Midgley et al., 2024). Recent breakthroughs in computer vision using compact latent spaces have achieved high sampling quality (Rombach et al., 2022), suggesting the potential of analogous approaches in the molecular domain.

## 4 EXPERIMENTS

### 4.1 LAGRANGIAN FLUID SIMULATION

We conducted experiments on two material point method (MPM) (Sulsky et al., 1995) datasets for our 2D experiments, namely WaterDrop and WaterDrop-XL from Sanchez-Gonzalez et al. (2020), which consist of a maximum of 1.1k and 7k particles, respectively, see Appendix A.4. Additionally, we introduce a new dataset of 3D dam break simulations called DamBreak3D generated using the

Riemann SPH method (Zhang et al., 2017) and the SPHinXsys library (Zhang et al., 2021). This dataset has between 145k-215k fluid particles and consists of 800/100/100 trajectories of length 250 steps, obtained after temporal subsampling at every 100th SPH step, more details in Appendix A.7.

**Metrics**. We evaluate the performance of the models in terms of 1) the intersection over union $IoU$ of occupancies and 2) the velocity MSE denoted $MSE$. To compute these metrics, we evaluate both occupancies and velocities on a regular grid spanning the full computational domain with a spacing of around twice the average particle spacing. While these grid values can be directly evaluated by querying the UPT++ decoder, we apply an SPH interpolation to compute them from the dataset and also for the GNN baselines.

**GNNs for large particle systems**. As our main baseline, we choose the established particle-based fluid mechanics surrogate GNS introduced by Sanchez-Gonzalez et al. (2020). However, as seen in Figure 3, such GNN-based approaches do not scale well. To the best of our knowledge, there are three main directions for scaling GNNs to larger particle systems: A) evaluating subgraphs and combining the solutions (Bonnet et al., 2022), B) limiting the receptive field of the neural network and applying domain decomposition (Musaelian et al., 2023; Kozinsky et al., 2023), and C) using a hierarchy of coarser graphs (Qi et al., 2017; Fortunato et al., 2022; Lino et al., 2022). Regarding A, to cover a mesh with 150k nodes, AirFRANS (Bonnet et al., 2022) evaluates 100 randomly sampled subgraphs of 32k nodes covering the whole domain and averages the outputs on the nodes that have been evaluated multiple times. Regarding B, Allegro (Musaelian et al., 2023) proposes a novel paradigm which, in contrast to message passing, operates on strictly local neighborhoods to allow for straightforward domain decomposition. Although Allegro allows for simulating systems of arbitrary size, the compute requirement scales linearly with the system size – in a scaling example, Allegro distributes 100M atoms over 5k GPUs or roughly 20k atoms per GPU (Kozinsky et al., 2023). Thus, both A and B approaches have a linear or worse scaling of compute with respect to system size. As we aim to develop a framework that scales to at least 200k particles (in our experiments), the hierarchical approach C seems most suitable. Thus, to have a competitive baseline, we develop a multi-scale version of GNS, called MS-GNS, which couples the finer (original) particles with coarser particles consisting of randomly subsampled 12.5% of the finer particles. Our approach is inspired by MS-MGN (Fortunato et al., 2022) and connects the two point clouds by a k-nearest neighbors graph with $k = 4$ from the fine to the coarse nodes, see Appendix A.5 for more details.

**Model details**. In our experiments, UPT++ encodes the first two velocities of the trajectory and the time-evolution operator acts only in the latent space. In contrast, GNS encodes the first five velocities and then autoregressively predicts and integrates the accelerations. Based on the GNS ablation studies in Sanchez-Gonzalez et al. (2020) and Toshev et al. (2023b), we choose to train a model with 10 message-passing (MP) steps and a latent size of 128, denoted by GNS-10-128 in Table 1. With MS-GNS-15, we denote an MS-GNS model with a processor consisting of: 1 MP layer on the fine particles with a latent size 64, 1 downsampling layer, 11 MP layers on the coarse graph with a latent size 128, 1 upsampling layer, and 1 MP layer as the first one.

**Results**. Our main results are summarized in Table 1, showing that UPT++ performs comparably to GNNs in terms of both $IoU$ and $MSE$, but can offer more than 50x greater speedups. Notably, we work with the smaller GNS-5-64 model on DamBreak3D as this is the biggest GNS model we could train with one sample per 40GB GPU, compare Figure 3. Qualitatively, the lower $IoU$ of UPT++ on the 2D datasets is related to the lack of mass conversation (equivalently volume conservation, as we work with incompressible fluids) in the latent state representation, which manifests itself in having too little or too much fluid along a trajectory. We note that volume conservation is a problem that GNNs also have, as recently discussed in Neural SPH (Toshev et al., 2024), but in contrast to UPT++, GNNs, by construction, preserve the mass, i.e. the number of particles. On the other hand, UPT++ learns a better representation of the velocity field, which we suspect is easier to learn as a continuous function. Figure 1 shows one exemplary trajectory rollout of DamBreak3D, demonstrating that UPT++ can adequately model a 3D particle simulation with 200k particles.

## 4.2 SAMPLING MOLECULAR CONFORMATIONS

We apply UPT++ to the task of sampling molecular conformations. Molecular conformations are, e.g., important when studying interactions between different molecules or when deriving certain

Table 1: MSE denotes the mean-squared error of the velocity prediction. IoU and MSE are averaged over all timesteps and all trajectories in the test set of each dataset. UPT++ can model the complex dynamics while providing a significant speedup.

| Dataset | Method | Hardware | Rollout time | Speedup | $IoU$ ($\uparrow$) | $MSE$ ($\downarrow$) |
|---|---|---|---|---|---|---|
| WaterDrop | MPM | 6 CPUs | 50s | 1x | - | - |
| | GNS-10-128 | A40 | 4.0s | 13x | **0.91** | 0.047 |
| | UPT++ | A40 | **0.53s** | **94x** | 0.87 | **0.036** |
| WaterDrop-XL | MPM | 6 CPUs | 170s | 1x | - | - |
| | GNS-10-128 | A40 | 44s | 4x | **0.83** | **0.16** |
| | UPT++ | A40 | **0.65s** | **262x** | 0.81 | **0.16** |
| DamBreak3D | SPH | 32 CPUs | 1200s | 1x | - | - |
| | GNS-5-64 | A40 | 100s | 12x | 0.83 | 0.54 |
| | MS-GNS-15 | A6000 | 150s | 8x | 0.91 | 0.18 |
| | UPT++ | A40 | **2.7s** | **444x** | **0.93** | **0.14** |

properties for molecules. This might be especially relevant for biomedical chemistry and drug discovery.

We benchmark UPT++ on two small peptide datasets from Klein et al. (2024a), namely the alanine dipeptide dataset (*AD*), containing a single molecule of 22 atoms, and, further on a small peptide dataset (*2AA*) containing 400 different peptides, with varying number of atoms (20-50) and different atom types per molecules. We stick to the suggested train/test split provided with *2AA* and either evaluate on the whole test set or on an exemplary molecule (*AN*) from the test set.

**Metrics**. We evaluate the performance of UPT++, implemented via a latent sampling operator, by investigating associated Ramachandran plots (Ramachandran et al., 1963) for the sampled molecule conformations. These plots show the distribution of peptide dihedral angles $\phi$ and $\psi$. We quantify the differences of marginalized distributions of angles between our sampled molecule conformations and those ones from a reference simulation in analogy to Yu et al. (2024) by means of the Jenson-Shannon divergence.

**Model specific details and training details**. We use a guided flow-matching model (as explained above) with the same diffusion transformer backbone (Peebles & Xie, 2023) as for the time-evolution operator. As outlined above, we do not only condition on a previous molecule conformation at time $t$, but also on the number of atomistic timesteps $N$. The idea is partly based on ITO (Schreiner et al., 2023). Further, we apply an MD relaxation step before reconditioning (see Appendix B.2.3). Details on importance-based sampling for molecules and on data augmentation we use can be found in Appendices B.2.2 and B.2.4.

**Results**. Figure 6 shows Ramachandran plots and free energy projection plots for *AD* conformations for 13.6k UPT++ sampled conformations and 800k MD reference simulation conformations, respectively. The Ramachandran plots indicate that modes of reference conformation angles are faithfully restored by sampled UPT++ conformations. The free energy projections show that our model also captures energy minima very well, and that less likely regions in conformation space are explored. For exemplary molecules (*AN*) from the test set of *2AA* Ramachandran plots and free energy projection plots are shown in Figures B2 and B3 in Appendix B.5. All *AN* plots rely on 10k UPT++ sampled conformations and 9.8k MD reference simulation conformations. We investigate the influence of the flow-matching guidance parameter and observe that less guidance seems to capture modes of conformation angles better than high guidance. This is also reflected by the Jensen–Shannon divergences shown in Table 2.

**Extracting molecule graphs after decoding signals from latent space**. To assess if molecule graph extraction from the latent space happens uniquely, we evaluate the graph extraction performance of UPT++ in two scenarios: first, as a strict autoencoder, i.e. encoding and decoding without the sampling operator, and second, with the sampling operator applied. We consider the extraction (for details on the final graph extraction see Appendix B.4) of a molecule as valid when encoded and decoded versions have equal InChI (Landrum, 2016) codes, utilizing the RDKit library (Karol, 2018). This ensures chemical identity is preserved. Table 2 shows that the molecule encoder and

Table 2: Summary of results on molecular sampling. We show the reconstruction success rate without applying the sampling operator ("enc-dec"), and with sampling ("sampling+dec").

| Dataset | Guidance scale | Reconstr. success rate (↑) | | JS Ramachandran (↓) (sampling) |
|---------|------|-------|-------|-------|
|  |  | enc-dec | sampling + dec |  |
| AD | 3.5 | 1.0 | 0.97 | 0.18 |
| 2AA: AN | 1.0 | 1.0 | 0.20 | 0.30 |
| 2AA: AN | 3.5 | 1.0 | 0.50 | 0.17 |

decoder of UPT++ can accurately extract molecules from the latent space for Alanine dipeptide and all test molecules from the 2AA dataset (success rate 100% when tested with 1000 samples). When applying the sampling operator, the extraction performance decreases slightly (0.97). Results for the AN molecule of the 2AA dataset are worse, which results from the fact that the tested molecules are unseen during training. Further, higher guidance scales increase reconstruction success rates but yield less diverse samples. This behavior is analogous to classifier-free guidance in other domains.

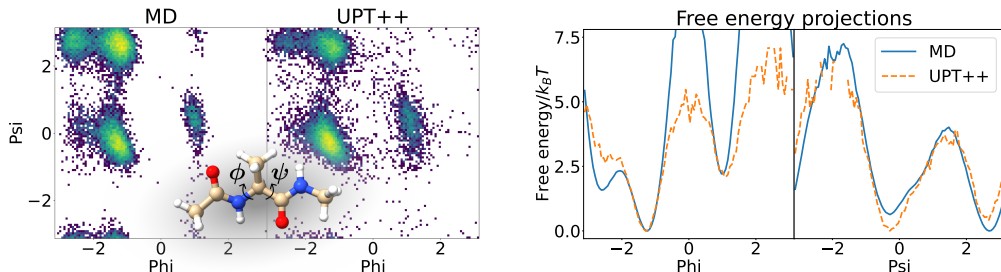

Figure 6: **Alanine dipeptide experiments.** *Left half:* Ramachandran plots comparing 14k UPT++ and MD samples. *Right half:* Free energy surface for the two dihedral angles $\psi$ and $\phi$ with the same 14k UPT++ samples but 800k MD samples. Our model captures well the energy minima and also explores less probable regions of sample space.

## 5 CONCLUSION, LIMITATIONS AND FUTURE WORK

We presented UPT++, a generic framework for modeling simulations that are conventionally discretized with particles. UPT++ maps the state of the system to a fixed-sized latent space with novel importance-based encoding and importance-based decoding techniques. The strengths of our approach are its generic architecture, favorable scaling to larger systems, and potential for significant acceleration of numerical simulations. We demonstrated these strengths by training a Lagrangian fluid dynamics surrogate on up to 200k particles, as well as on molecular conformer generation across different peptide molecules.

Amongst others, possible extensions of UPT++ are adoptions to more challenging engineering problems, e.g., complex geometries, solid-liquid interactions, or multi-physics. The fact that the size of latent representations is constant in UPT++ and, therefore, in principle independent of concrete molecule sizes, suggests its application to larger peptides or other larger molecules. One weakness of a field-based approach like UPT++ is that – by construction – UPT++ is not mass conserving. This is in contrast to GNNs, which preserve the mass, i.e., the number of particles. Further, for molecular conformation sampling, we decided to incorporate the conditioning conformations via a classifier-free guidance approach since directly conditioning on previous conformations is problematic, as discussed in literature (Li et al., 2024). We leave the exploration of other conditioning strategies to future work. Lastly, improving the decoding scheme could significantly speed up conformer sampling.

## ETHICS STATEMENT

**Accuracy and Reliability of Simulations**. While UPT++ offers a computationally efficient alternative to traditional simulation methods, there is a risk that inaccuracies in the approximations could lead to unintended consequences. For example, in a civil engineering context, designs for flood protection could rely significantly on the accuracy of our simulations. We strongly emphasize that our models should not be blindly relied upon for decision-making in safety-critical areas. Users should always corroborate machine learning predictions with established physical models or additional empirical data. Similarly for molecular simulations, the output of our models should be checked with experiments or complemented with classical physical simulations before decision-making.

**Transparency and Explainability**. Given that UPT++ encodes the system's state into a continuous latent representation, it may offer less transparency and explainability compared to methods that directly operate on the physical state. Our latent space approach can make it difficult for users to fully understand how certain predictions or decisions are reached. Lack of interpretability could lead to challenges in trusting and verifying the model's outputs, particularly in critical applications. To mitigate these concerns, we advocate for developing methods that enhance explainability and allow users to inspect and understand the underlying decision processes of UPT++, ensuring its safe and responsible use in real-world scenarios.

**Environmental and Social Impact**. Our models could have significant societal and environmental impacts. For example, in cases like flood prediction or water resource management, inaccurate predictions may lead to poor planning or resource allocation, disproportionately affecting vulnerable communities. We urge that such models be used responsibly, with attention to fairness, inclusivity, and transparency, especially in areas that affect public health, safety, and well-being.

## REPRODUCIBILITY STATEMENT

We provide a detailed description of the model in Section A.3 and B.2, along with a comprehensive table for the hyperparameters used in Lagrangian fluid simulations in Table A1 and the hyperparameters used in sampling molecular conformations in Table B1. The data for experiments conducted on WaterDrop and WaterDrop-XL can be found in Sanchez-Gonzalez et al. (2020). The DamBreak3D dataset will be released soon. The data for experiments conducted on the MD experiments can be found in Klein et al. (2024a). Additionally, we provide the anonymized code in the supplementary materials. The finalized code, along with all model checkpoints used in the experiments, will be made publicly available on GitHub.

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

CONTENTS

# A LAGRANGIAN FLUID SIMULATION

## A.1 SMOOTHED PARTICLE HYDRODYNAMICS (SPH)

In contrast to Eulerian approaches, where discretization of the continuous space is achieved through spatially fixed finite nodes, control volumes, cells, or elements, Lagrangian methods employ finite material points, often termed *particles*, whose movement aligns with the local deformation of the continuum. One of the most prominent Lagrangian discretization schemes is smoothed particle hydrodynamics (SPH), originally proposed by Lucy (1977) and Gingold & Monaghan (1977) for applications in astrophysics. SPH approximates the field properties using radial kernel interpolations over adjacent particles at the location of each particle. The strength of the SPH method is that it does not require connectivity constraints, e.g., meshes, which is particularly useful for simulating systems with large deformations. Since its foundation, SPH has been greatly extended and is the preferred method to simulate problems with (a) free surfaces (Marrone et al., 2011; Violeau & Rogers, 2016), (b) complex boundaries (Adami et al., 2012), (c) multi-phase flows (Hu & Adams, 2007), and (d) fluid-structure interactions (Antoci et al., 2007). SPH approximates the incompressible Navier-Stokes equations (NSE) by the so-called weakly compressible NSE, where the weak compressibility assumption typically allows for up to $\sim 1\%$ density deviation Monaghan (2005). This $\sim 1\%$ is enforced for the weakly compressible SPH method while evolving density and momentum:

$$\frac{\mathrm{d}}{\mathrm{d}t}(\rho) = -\rho\left(\nabla \cdot \boldsymbol{u}\right), \tag{A.1}$$

$$\frac{\mathrm{d}}{\mathrm{d}t}(\boldsymbol{u}) = \underbrace{-\frac{1}{\rho}\nabla p}_{\text{pressure}} + \underbrace{\frac{1}{Re}\nabla^2\boldsymbol{u}}_{\text{viscosity}} + \underbrace{\boldsymbol{f}}_{\text{ext. force}}. \tag{A.2}$$

Herein, $\rho$ is the density, $\mathbf{u}$ the velocity, $p$ the pressure, $\boldsymbol{f}$ an external force, $Re \propto 1/\mu$ the Reynolds number. Solving these equations with standard SPH methods may still produce artifacts, most notably when particle clumping exceeds the 1% density-fluctuation restriction (Adami et al., 2013; Toshev et al., 2024).

The Material Point Method (MPM) is another particle-based technique that represents material as an assembly of material points. The motion of each material point is determined by solving Newton's laws of motion. MPM adopts a hybrid Eulerian-Lagrangian scheme, which uses moving material points and a fixed computational grid. MPM is particularly useful in the context of large deformations including fracture and contact scenarios, where traditional mesh-based methods might yield unrealistic or undesired outcomes due to mesh distortions.

## A.2 SCALING LIMITS

In Figure 3, we compare the memory consumption between UPT++, GNS-10-128, and GNS-5-64 during training. We construct a toy setting based on DamBreak3D, positioning points on a regular three-dimensional grid with the same particle spacing $\Delta x$ used in DamBreak3D (see Section A.7 for details). We start with a grid of 16x16x16 points and double the last dimension repeatedly, increasing the size from 16x16x16 to 16x16x8192. This results in configurations ranging from 4k points to over 2 million points. For UPT++, we scale the number of input points selected, the number of supernodes $n_S$, and the number of points where we decode the velocity and the occupancy based on the number of particles. We select $25\%$ of the points as input points, use $2.5\%$ as supernodes, decode the velocity at $10\%$ and decode the occupancy at $20\%$, which is similar to our setting used for DamBreak3D, compare Section A1). We fix the number of latent tokens to $n_{latent} = 4096$. We focus on memory consumption during the first training stage, where the encoder and decoder are trained, as the second stage requires less memory.

The main memory consumption that can be attributed to UPT++ is the query MLP in the decoder, which scales linearly with the number of points to decode, both for decoding the occupancy and the velocity. We can further reduce the memory footprint by decoding a smaller fraction of the points, as can be seen in Figure A1, where we add two scaling plots where we decode only $5\%$ and only $1\%$ of the points.

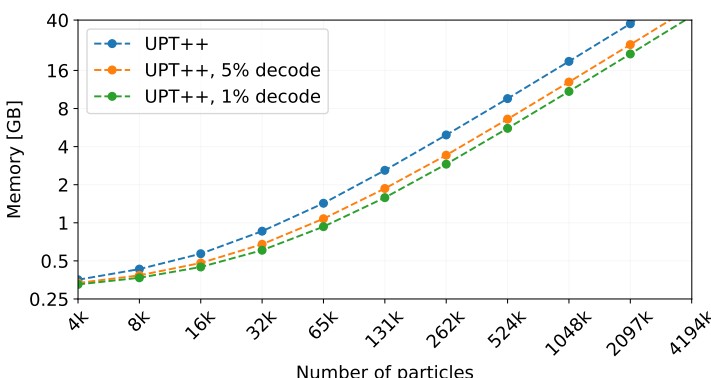

Figure A1: Memory usage of UPT++ variants with a reduced number of decoding points.

## A.3 IMPLEMENTATION DETAILS

The following outlines the implementation details for UPT++. Table A1 gives a detailed overview of the hyperparameters used in training. If there is a change in the dimensions between different blocks, we perform a learnable linear projection. All transformer (Vaswani et al., 2017) and perceiver (Jaegle et al., 2021b) blocks use standard pre-norm architecture as used in ViT Dosovitskiy et al. (2020) and are modulated by the timestep using DiT modulation (Peebles & Xie, 2023). We use layer normalization Ba et al. (2016) in the output of the encoder as well as in the input and output of the forward operator to always keep the latent representation normalized.

All experiments use the AdamW optimizer (Kingma & Ba, 2015; Loshchilov & Hutter, 2019) and follow a learning rate schedule that begins with a linear warmup and transitions to cosine decay (Loshchilov & Hutter, 2017). We perform early stopping and use the best checkpoint in terms of $IoU$ evaluated on the validation set of each dataset respectively.

**Encoder**.  First we sample a subset out of all particles and project two consecutive velocities into a higher dimension using a linear layer. Then we sample $n_S$ supernodes from the input point cloud and perform message passing to points within a radius $r_S$ to process the local information. In message passing, the features of the supernode and the point are concatenated along with a positional embedding that captures their relative distance. The supernode features are then processed by a transformer to capture global information. To further reduce the number of tokens in the latent space, we apply perceiver pooling with learned queries, followed by layer normalization, producing a normalized latent representation.

**Latent space operator**.  We apply layer normalization to the latent representation before passing it into the transformer blocks. The output is then added to the original latent representation (after normalization) and passed through another layer normalization step, ensuring the latent representation remains consistently normalized throughout the process. The timestep conditioning of the transformer blocks uses the timestep of the latent representation in the input.

**Decoder**.  The decoder takes the latent representation and processes it with a small Transformer, which is then fed into two perceiver blocks, one for decoding the state and one for decoding the occupancy. The positions where we want to query the latent representation are transformed into a positional embedding and fed through an MLP, resulting in the query used by the perceiver. The keys and values are the outputs of the Transformer, and the result of the perceiver is projected into the input dimension of the physical state or into a two-dimensional output for the occupancy.

**Timestep modulation**.  To incorporate information from the timestep, we encode the current timestep into a positional embedding. We use the transformer positional encoding Vaswani et al. (2017); Gupta & Brandstetter (2022), and, therefore rescale all timesteps to the range [0, 200]. The resulting timestep embedding is then used to perform DiT modulation (Peebles & Xie, 2023) for all transformer and perceiver blocks. DiT modulation involves applying dimension-wise scaling, shifting, and gating operations to both the attention and MLP modules of the transformer and perceiver blocks.

Table A1: UPT++ hyperparameters for the application to Navier-Stokes equations.

| Hyperparameter | WaterDrop | WaterDrop-XL | DamBreak3D |
|---|---|---|---|
| **General model parameters** | | | |
| Number of latent tokens $n_{\text{latent}}$ | 128 | 128 | 512 |
| Timestep embedding dim | 192 | 192 | 192 |
| DiT conditioning dim | 768 | 768 | 768 |
| **Encoder** | | | |
| Range of input points selected $k$ | 400 - 800 | 1k-3k | 32k-64k |
| Input features | 4 | 4 | 6 |
| Node features | 96 | 96 | 96 |
| Num. supernodes $n_{\text{S}}$ | 128 | 512 | 4096 |
| Supernode radius $r_{\text{S}}$ | 0.05 | 0.05 | 0.15 |
| Max supernode neighbours | 8 | 8 | 32 |
| Relative positional embedding dim | 96 | 96 | 96 |
| Message passing MLP dims | 288/96 | 288/96 | 288/96 |
| Transformer dim / layers / heads | 96/4/2 | 96/4/2 | 96/4/2 |
| Perceiver dim / num heads | 192/3 | 192/3 | 192/3 |
| **Forward operator** | | | |
| Transformer dim / layers / heads | 192/12/3 | 192/12/3 | 192/12/3 |
| **Decoder** | | | |
| Transformer dim / layers / heads | 192/4/3 | 192/4/3 | 192/4/3 |
| Query MLP dims | 768/768/192 | 768/768/192 | 768/768/192 |
| Perceiver dim / num heads | 192/3 | 192/3 | 192/3 |
| Output features | 4 | 4 | 6 |
| Number of points to decode (velocity) $k'$ | 125 | 500 | 16k |
| Number of points to decode (occupancy) $k'_{\text{o}}$ | 250 | 1000 | 32k |
| Occupancy radius of a particle | 0.01 | 0.01 | 0.05 |
| **First training stage** | | | |
| Num. epochs | 10 | 10 | 10 |
| Learning rate | 5e-3 | 5e-4 | 5e-4 |
| Weight decay rate | 0.05 | 0.05 | 0.05 |
| Warmup epochs | 2 | 2 | 2 |
| Batch size | 1024 | 256 | 32 |
| **Second training stage** | | | |
| Num. epochs | 10 | 10 | 10 |
| Learning rate | 5e-4 | 5e-4 | 5e-4 |
| Weight decay rate | 0.05 | 0.05 | 0.05 |
| Warmup epochs | 2 | 2 | 2 |
| Batch size | 256 | 256 | 64 |

**Training**. In the first training stage, we train both the encoder and decoder without the time-evolution operator. We do this by sampling states from a trajectory at timestep $t$, selecting $k$ input points, decoding at $k'$ output points, and regressing the velocity at these points using an MSE objective. Additionally, we randomly sample $k'_{\text{o}}$ points within the domain to obtain the ground truth occupancy. If a coordinate $x$ lies outside the occupancy radius of all particles, it is labeled as unoccupied; otherwise, it is marked as occupied by the fluid. We train using a CE loss. In the second training stage, we freeze the encoder and encode two consecutive timesteps, $t$ and $t'$, into latent space representations $z^t$ and $z^{t'}$. The time-evolution operator uses $z^t$ as input to predict $z^{t'}$, and we train the time-evolution operator using an MSE objective.

## A.4 DATASET-SPECIFIC DETAILS

In the following Table A2, we summarize the size of the used datasets, emphasizing that we work with a median number of particles all the way from 500 through 4,000 to 180,000. Each dataset consists of 1000 trajectories in the training set and 100 trajectories in both the validation and test sets.

Table A2: Statistics of particle counts and trajectory length in our Lagrangian fluid dynamics datasets.

| Dataset | number of particles | | | Trajectory length |
|---------|------|--------|------|-------------------|
| | min | median | max | |
| WaterDrop | 195 | 548 | 1,108 | 1000 |
| WaterDrop-XL | 1,948 | 4,031 | 7,184 | 1000 |
| DamBreak3D | 144,133 | 179,312 | 215,661 | 250 |

## A.5 BASELINES

Our baselines on the Lagrangian fluid dynamics problems are GNS (Sanchez-Gonzalez et al., 2020) and its multi-scale version MS-GNS. We adopt the Pytorch implementation of GNS from github.com/wu375/simple-physics-simulator-pytorch-geometry to our codebase and reuse most building block in MS-GNS. In the following, we provide more details about MS-GNS and the training hyperparameters.

**MS-GNS**. To the best of our knowledge, our baseline model MS-GNS is the first multi-scale GNN for Lagrangian fluid dynamics, and it combines ideas from Fortunato et al. (2022) and our own importance-based encoder approach. We acknowledge that there are various multi-scale GNNs that operate on static objects like point sets (Qi et al., 2017; Lino et al., 2022) or meshes (Fortunato et al., 2022; Suk et al., 2023), but because these discretizations are static, one can precompute coarser versions thereof using various different algorithms. However, in Lagrangian numerical methods, we need to compute a coarse graph at every timestep of the autoregressive rollout evolution, making advanced algorithms like furthest point sampling (FPS) (Qi et al., 2017) unfeasible – see tested FPS on DamBreak3D –, taking 10x more time than the model forward evaluation. Thus, for constructing the coarser graph, we resort to what we do in the sampling-based UPT++ encoding, namely randomly picking a subset of the nodes. The only difference to the encoder is that we do not have a fixed number of supernodes, but rather a relative ratio of subsampled nodes, which we set to $0.5^{dim}$, which essentially means that we go to particles with 2x the radius of the finer particles, which also means that we can just double the cutoff radius for the coarser graph. The obvious disadvantage of this method is that some fine particles might be far away from the coarser particles, which we remedy by constructing the mapping from finer to coarser graph using k-nearest neighbors with $k = 4$ – 4 basically means that a fine particle sees either its corresponding coarse particle and 3 others, or just 4 coarse particles. This way every fine node is guaranteed to get access to information from the coarser nodes, and because we subsample the coarse nodes anew at every timestep, the information propagation is well distributed. Other than the coarse graph generation and the fine-coarse mapping graph, our approach is almost equivalent to MS-MGN by Fortunato et al. (2022), with the only difference being that all latent vectors connected with the original finely resolved graph have half the size of the latent vectors of the coarse graph; this adjustment significantly reduces the memory of the forward pass. Overall, the message-passing steps that operate only on the fine or coarse graphs are exactly the GNS layers, and the mapping between the resolutions happens along the same k-NN graph.

The hyperparameter setting for MS-GNS is one of what Fortunato et al. (2022) found to work well, namely a simple V-shaped processor scheme which we call MS-GNS-15 consisting of: 1 MP layer on the fine scale, downsampling layer, 11 MP layers on the coarse graph, upsampling layer, and 1 more MP layer on the fine graph. Regarding the training protocol, we train GNS and MS-GNS with the same optimizer and learning rate scheduler as our other models. A summary of the hyperparameters used for training the GNN baselines is given in Table A3, which complements Appendix A.3.

Table A3: GNN hyperparameters overview.

| Hyperparameter | GNS-10-128/GNS-5-64 | MS-GNS-15 |
|---|---|---|
| **Physical input/output features** | | |
| Num. input velocities | 5 | 5 |
| Node input features | velocity, boundary dist. | velocity, boundary dist. |
| Edge input features | displacement | displacement |
| Node output features | acceleration | acceleration |
| Include magnitudes | yes | yes |
| **GNN architecture** | | |
| MP layers (if appl. fine) | 10/5 | 2 |
| Upsampling layers | - | 1 |
| Downsampling layers | - | 1 |
| MP layers coarse | - | 11 |
| Latent dimension (if appl. fine) | 128/64 | 64 |
| Latent dimension coarse | - | 128 |
| Num. MLP layers | 2 | 2 |
| Noise std | 6.7e-4 | 6.7e-4 |
| **Training configuration** | | |
| Num. epochs | 10 | 10 |
| Learning rate | 1e-4 | 1e-4 |
| Weight decay rate | 0.05 | 0.05 |
| Warmup epochs | 2 | 2 |
| Batch size | {WaterDrop: 2, WaterDrop-XL: 10, DamBreak3D: 4} | |

## A.6 ADDITIONAL RESULTS

In Figure A2 we present the IoU and the velocity error for full rollouts on the test set. At the start of the trajectory, GNS has an advantage by numerically integrating the accelerations. However, during the highly dynamic phase between timesteps 200 and 500, the difference becomes negligible. Figure A3 and A4 illustrate this by presenting snapshots of a trajectory rollout for both Waterdrop and Waterdrop-XL. Similarly, as demonstrated for DamBreak3D in Figure 1, UPT++ captures the overall fluid dynamics in both the smaller Waterdrop and Waterdrop-XL scenarios. In the end of each trajectory, as the fluid settles at the bottom of the box, GNS inherently preserves mass, or the number of particles, which enables it to more accurately capture the volume. In contrast, UPT++ latent propagation lacks such constraints, making it unable to accurately capture the fluid's volume.

## A.7 DAM BREAK 3D DATASET

We generated the dataset by modifying the 3D dam break test case in the SPHinXsys library (Zhang et al., 2021)[1]. In particular, we modify a) the numerical integration scheme and b) the initial geometry of the fluid.

Regarding the integrator, we modify the adaptive-step dual-criteria time stepping scheme (Zhang et al., 2020) by fixing the step size $\Delta t_{inner} = 0.0008$ of the inner pressure and density relaxation loop and also by fixing the number of iterations in this inner loop to 5. The reason for this is that we want equidistantly spaced samples in time to not have to deal with conditioning on the timestep size in the ML problem formulation. The chosen $\Delta t_{inner}$ is close to the worst-case adaptively estimated one but still does not significantly change the number of integration steps to reach the end time of 20. Note that we omit units here as the simulation is of the non-dimensionalized NSE. With the temporal coarsening level of 100 relative to the inner loop steps, each simulation has $20/(0.0008 \cdot 100) = 250$ steps (to be more precise, 251 steps, as we record both the very first and last states).

Regarding the parametrization of the geometry, we sample 12 random numbers determining the shape of the top and front of the wave, and after the fluid volume is filled with particles, we add

---

[1]https://github.com/Xiangyu-Hu/SPHinXsys

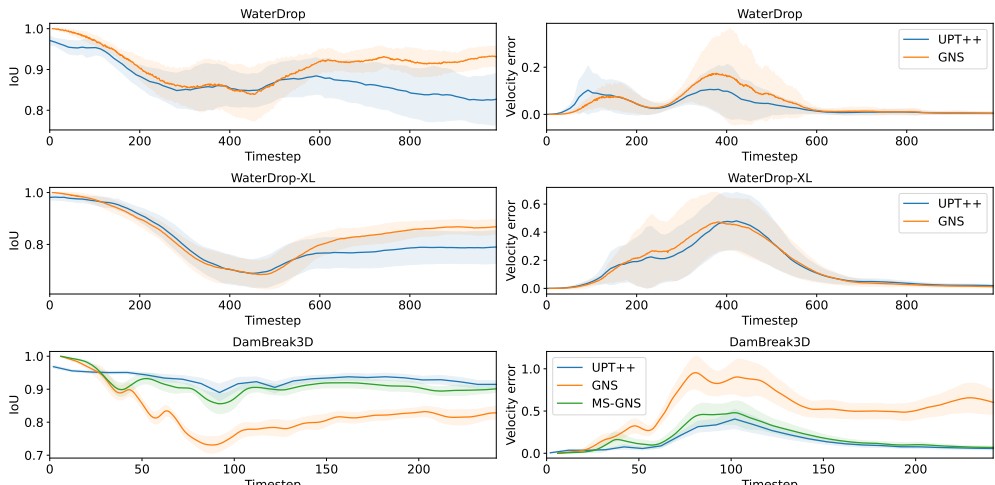

Figure A2: Mean and standard deviation of the IoU and the velocity error during the rollout of all trajectories in the test set. For WaterDrop and WaterDrop-XL, the first simulation steps are better predicted by GNS, which is expected because GNS numerically integrates accelerations, and also, the last steps, where the fluid is resting at the bottom, are better predicted. During the highly dynamic part, GNS and UPT++ are on par, while UPT++ better predicts the correct velocity. For DamBreak3D, GNS is not able to predict the rollout of the large-scale trajectory, but UPT++ and MS-GNS can handle this task.

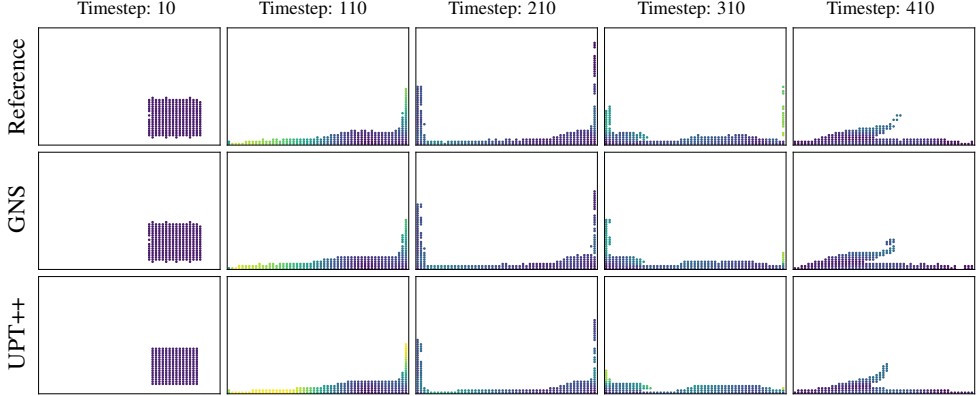

Figure A3: Various timesteps along the WaterDrop trajectory are evaluated on a regular grid for comparison purposes. The presence of a point on the grid represents its occupancy, while its color indicates the magnitude of the velocity.

Gaussian noise to the coordinates. The standard deviation of the noise is $\sigma = 0.1 \cdot \Delta x$ with $\Delta x = 0.025$ being the particle spacing. The computational domain begins at $(0, 0, 0)$ and spans $L \times H \times W = 5.366 \times 2 \times 2$, with the number 5.366 coming from the original dam break experiments by Colagrossi & Maurizio (2003). The fluid always fills the bottom left part of the domain (at $x = 0, y = 0$) spanning the full width, and we modulate the top and front sides by the mentioned 12 numbers defining sinusoidal waves by their amplitude $a$, period $p$, and shift $s$. The top surface of the fluid is defined by its height $h_{top}(x, z)$ as a function of the length and width ($x$ and $z$ axes), and the x-coordinate (length) of the front $l_{front}(z, y)$ is defined as a function of the width and height.

$$h_{top}(x, z) = H_{ave} + a_{top,x} \cdot \sin\left(2\pi(p_{top,x} \cdot x/L_{ave} + s_{top,x})\right)$$
$$+ a_{top,z} \cdot \sin\left(2\pi(p_{top,z} \cdot z/W_{ave} + s_{top,z})\right)$$
$$l_{front}(z, y) = L_{ave} + a_{front,z} \cdot \sin\left(2\pi(p_{front,z} \cdot z/W_{ave} + s_{front,z})\right)$$
$$+ a_{front,y} \cdot \sin\left(2\pi(p_{front,y} \cdot y/H_{ave} + s_{front,y})\right)$$

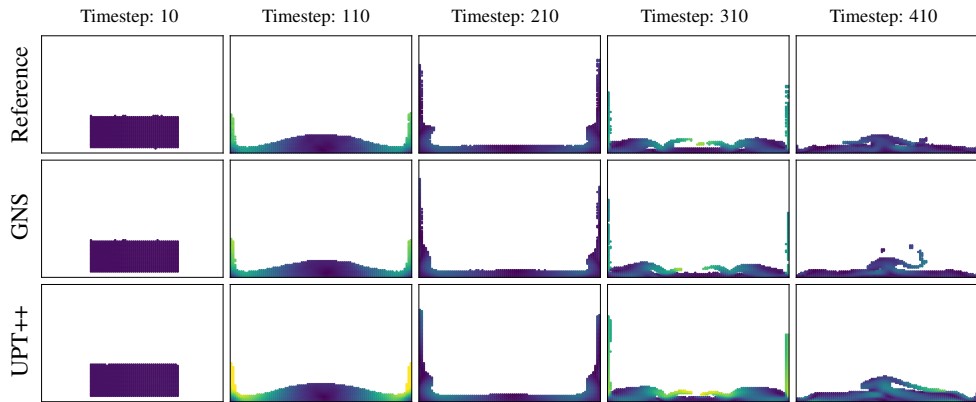

Figure A4: Various timesteps along the WaterDrop-XL trajectory are evaluated on a regular grid for comparison purposes. The presence of a point on the grid represents its occupancy, while its color indicates the magnitude of the velocity.

The values of the average length, height, and width of the fluid are $L_{ave} = 2$, $H_{ave} = 0.7$, and $W_{ave} = 2$, respectively. The random numbers for $a, p, s$ are sampled uniformly from $a \sim \mathcal{U}(0, 0.15)$, $p \sim \mathcal{U}(0.25, 2)$, $s \sim \mathcal{U}(0, 1)$. We visualize the first 10 trajectories from the train split in Figure A5.

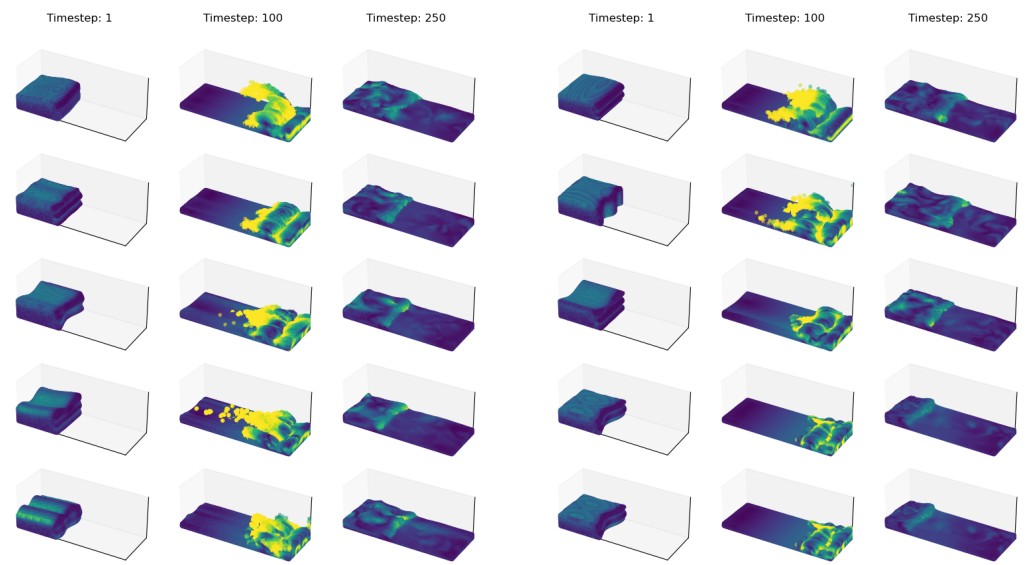

Figure A5: Frames 1, 100, and 250 from the first 10 training trajectories of DamBreak3D. The color is the velocity magnitude estimated by subtracting the previous positions from the current ones.

# B   MOLECULAR CONFORMATION SAMPLING

## B.1   MOLECULAR DYNAMICS (MD)

The most fundamental concepts nowadays to describe the dynamics of molecules are given by the laws of quantum mechanics. The Schrödinger equation is a partial differential equation, that gives the evolution of the complex-valued wave function $\psi$ over time $t$: $i\hbar\frac{\partial\psi}{\partial t} = \hat{H}(t)\psi$. Here $i$ is the imaginary unit with $i^2 = -1$, $\hbar$ is reduced Planck constant, and, $\hat{H}(t)$ is the Hamiltonian operator at time $t$, which is applied to a function $\psi$ and maps to another function. It determines how a quantum system evolves with time and its eigenvalues correspond to measurable energy values of the quantum system. The solution to Schrödinger's equation in the many-body case (particles $1, \ldots, N$) is the wave function $\psi(\mathbf{x}_1, \ldots, \mathbf{x}_N, t) : \bigtimes_{i=1}^{N} \mathbb{R}^3 \times \mathbb{R} \to \mathbb{C}$ which we abbreviate as $\psi(\{\mathbf{x}\}, t)$. It's the square modulus $|\psi(\{\mathbf{x}\}, t)|^2 = \psi^*(\{\mathbf{x}\}, t)\psi(\{\mathbf{x}\}, t)$ is usually interpreted as a probability density to measure the positions $\mathbf{x}_1, \ldots, \mathbf{x}_N$ at time $t$, whereby the normalization condition $\int \ldots \int |\psi(\{\mathbf{x}\}, t)|^2 \, d\mathbf{x}_1 \ldots d\mathbf{x}_N = 1$ holds for the wave function $\psi$.

Analytic solutions of $\psi$ for specific operators $\hat{H}(t)$ are hardly known and are only available for simple systems like free particles or hydrogen atoms. In contrast to that are proteins with many thousands of atoms. However, already for much smaller quantum systems approximations are needed. A famous example is the Born–Oppenheimer approximation, where the wave function of the multi-body system is decomposed into parts for heavier atom nuclei and the light-weight electrons, which usually move much faster. In this case, one obtains a Schrödinger equation for electron movement and another Schrödinger equation for nuclei movement. A much faster option than solving a second Schrödinger equation for the motion of the nuclei is to use the laws from classical Newtonian dynamics. The solution of the first Schrödinger equation defines an energy potential, which can be utilized to obtain forces $\mathbf{F}_i$ on the nuclei and to update nuclei positions according to Newton's equation of motion: $\mathbf{F}_i = m_i \, \ddot{\mathbf{q}}_i(t)$ (with $m_i$ being the mass of particle $i$ and $\mathbf{q}_i(t)$ describing the motion trajectory of particle $i$ over time $t$).

Additional complexity in studying molecule dynamics is introduced by environmental conditions surrounding molecules. Maybe the most important is temperature. For bio-molecules it is often of interest to assume that they are dissolved in water. To model temperature, a usual strategy is to assume a system of coupled harmonic oscillators to model a heat bath, from which Langevin dynamics can be derived (Ford et al., 1965; Zwanzig, 1973). The investigation of the relationship between quantum-mechanical modeling of heat baths and Langevin dynamics still seems to be a current research topic, where there there are different aspects like the coupling of the oscillators or Markovian properties when stochastic forces are introduced. For instance, Hoel & Szepessy (2019), studies how canonical quantum observables are approximated by molecular dynamics. This includes the definition of density operators, which behave according to the quantum Liouville-von Neumann equation.

The forces in molecules are usually given as the negative derivative of the (potential) energy: $\mathbf{F}_i = -\nabla E$. In the context of molecules, $E$ is usually assumed to be defined by a force field, which is a parameterized sum of intra- and intermolecular interaction terms. An example is the Amber force field (Ponder & Case, 2003; Case et al., 2024):

$$E = \sum_{\text{bonds } r} k_b(r - r_0)^2 + \sum_{\text{angles } \theta} k_\theta(\theta - \theta_0)^2 + \tag{B.1}$$

$$\sum_{\text{dihedrals } \phi} V_n(1 + cos(n\phi - \gamma)) + \sum_{i=1}^{N-1}\sum_{j=i+1}^{N} \left( \frac{A_{ij}}{R_{ij}^{12}} - \frac{B_{ij}}{R_{ij}^{6}} + \frac{q_i q_j}{\epsilon R_{ij}} \right)$$

Here $k_b, r_0, k_\theta, \theta_0, V_n, \gamma, A_{ij}, B_{ij}, \epsilon, q_i, q_j$ serve as force field parameters, which are found either empirically or which might be inspired by theory.

Newton's equations of motions for all particles under consideration form a system of ordinary differential equations (ODEs), to which different numeric integration schemes like Euler, Leapfrog, or, Verlet can be applied to obtain particle position trajectories for given initial positions and initial velocities. In case temperature is included, the resulting Langevin equations form a system of

stochastic differential equations (SDEs), and Langevin integrators can be used. It should be mentioned, that it is often necessary to use very small integration timesteps to avoid large approximation errors. This, however, increases the time needed to find new stable molecular configurations.

## B.2 IMPLEMENTATION DETAILS

We use the same implementation as outlined in Chapter A.3. The differences specific to the MD setting are explained below. Table B1 summarizes the hyperparameters used in the molecular sampling experiments.

Table B1: UPT++ hyperparameters for the application to molecular sampling.

| Hyperparameter | AD | 2AA |
|---|---|---|
| **General Model Parameters** | | |
| Number of latent tokens $n_{\text{latent}}$ | 32 | 64 |
| Timestep embedding dim | 192 | 192 |
| DiT conditioning dim | 768 | 768 |
| **Encoder** | | |
| Range of input points selected | 2.7k | 6k |
| Input features | 4 | 4 |
| Node features | 96 | 96 |
| Num. supernodes $n_{\text{S}}$ | 128 | 512 |
| Supernode radius $r_{\text{S}}$ | 0.05 | 0.05 |
| Max supernode neighbours | 8 | 8 |
| Relative positional embedding dim | 96 | 96 |
| Message passing MLP dims | 288/96 | 288/96 |
| Transformer dim / layers / heads | 96/4/2 | 96/4/2 |
| Perceiver dim / num heads | 192/3 | 192/3 |
| **Forward operator** | | |
| Transformer dim / layers / heads | 32/22/3 | 32/22/3 |
| **Decoder** | | |
| Transformer dim / layers / heads | 192/4/3 | 192/4/3 |
| Query MLP dims | 768/768/192 | 768/768/192 |
| Perceiver dim / num heads | 192/3 | 192/3 |
| Output features | 5 | 5 |
| Number of points to decode (velocity) | 125 | 500 |
| Number of points to decode (occupancy) | 250 | 1000 |
| **First stage training** | | |
| Num. epochs | 1.7k | 73 |
| Learning rate | 1e-4 | 1e-4 |
| Schedule | Cosine | - |
| Batch size | 1024 | 1024 |
| **Second stage training** | | |
| Num. epochs | 4.6k | 53 |
| Learning rate | 1e-4 | 1e-4 |
| Batch size | 2048 | 256 |

### B.2.1 DENSITY REPRESENTATION

We represent molecules as density fields, following an approach similar to Pinheiro et al. (2024) and Dumitrescu et al. (2024). Each atom is represented by a 3D Gaussian-like density (Orlando et al., 2022; Li et al., 2014)

$$D(d, r) = \exp\left(-\frac{d^2}{(0.93 \cdot r)^2}\right), \tag{B.2}$$

where $D$ is the fraction of occupied volume by an atom with radius $r$ at distance $d$ from its center. While different occupancy radii could be considered for various atom types, we use a uniform radius of $r = 0.5$ Å for all atom types. The signal of the field for atom type $a$, with $I_a$ being an index array of all atoms within the molecule corresponding to atom type $a$, is defined as:

$$u_a^t(\boldsymbol{x}) = 1 - \prod_{n=1}^{|I_a|} \left( 1 - D\left( \left\| \boldsymbol{x} - \boldsymbol{m}_{I_a[n]}^t \right\|, r_a \right) \right), \tag{B.3}$$

where $\boldsymbol{m}_{I_a[n]}^t$ is the center location of atom $I_a[n]$ at time $t$ and $r_a$ is the radius for atom type $a$. With that, we obtain one density field $u_a$ per atom type $a$ (with $a \in \{H, C, \ldots\}$ ) and the joint signal for all atom types can be summarized by a vector of density fields $\boldsymbol{u}^t(\boldsymbol{x})$:

$$\boldsymbol{u}^t(\boldsymbol{x}) = (u_H^t(\boldsymbol{x}), u_C^t(\boldsymbol{x}), \ldots). \tag{B.4}$$

### B.2.2 IMPORTANCE-BASED SAMPLING FOR MOLECULES

We sample points from the density field vector $\boldsymbol{u}^t$ by first sampling sets of $N_{IS}$ points ($\{\boldsymbol{x}_1, \ldots, \boldsymbol{x}_{N_{IS}}\}$), where each point $\boldsymbol{x}_i$ is from a normal distribution centered around one of the input molecule atom locations $\boldsymbol{m}_k^t$ at time $t$ (with $k \in \{1, \ldots, N_{atoms}\}$, where $N_{atoms}$ is the number of atoms for the considered molecule):

$$\boldsymbol{x}_i \sim \mathcal{N}(\boldsymbol{m}_k^t, \boldsymbol{\sigma}^2) \tag{B.5}$$

Then we compute the associated signal vectors corresponding to the sampled points $\boldsymbol{x}_i$, i.e., $\boldsymbol{u}^t(\boldsymbol{x}_i)$. We use $\sigma = 0.5$ Å in all experiments, and add global nodes at the initial atom positions. Additionally, we randomly sample points in the input space and compute their signal.

### B.2.3 REFINEMENT

In autoregressive sampling, errors accumulate across inference steps, causing out-of-distribution issues. To mitigate this, we employed the energy minimization procedure described in (Yu et al., 2024), implemented using OpenMM (Eastman et al., 2017).

### B.2.4 DATA AUGMENTATION

During training we apply random rotation, uniform between $[0, 2\pi)$ along the three Euler angles to each training sample.During training we apply random rotation, uniform between $[0, 2\pi)$ along the three Euler angles to each training sample.

### B.3 NEURAL SAMPLING OPERATOR

In accordance with literature (Lipman et al., 2022; Liu et al., 2022), we assume a flow $\Phi$ to be created via a parameterized $(\theta)$ vector field $v_\theta$:

$$\frac{d\Phi(s, \boldsymbol{z})}{ds} = v_\theta(s, \Phi(s, \boldsymbol{z}), z_{\text{cond}}, N_{\text{cond}})$$
$$\Phi(0, \boldsymbol{z}) = \boldsymbol{z}$$

Here $s$ serves as the diffusion time. We build upon the idea of classifier-free guidance (Ho & Salimans, 2022) for flow matching (Zheng et al., 2023) to incorporate previous molecule conformations as condition a $(z_{\text{cond}})$. We further build upon ITO (Schreiner et al., 2023) to predict for more than one atomistic time step into the future and therefore also condition on a number of time steps $(N_{\text{cond}})$. Algorithm 1 shows how $v_\theta$ can be trained given a series of MD trajectories. Algorithm 2 then shows how new samples can be generated using the trained flow matching velocity field $v_\theta$ and a given previous conformation state as well as the number of atomistic time steps. The flow matching guidance parameter $\omega$ and the employed number of ODE steps ($N_{\text{ODE}}$) serve as hyperparameters.

---

**Algorithm 1** Training UPT++ sampling operator

---

1: **Inputs:**
- $n_{\mathcal{Z}}$ MD-trajectories $\mathcal{Z} = \left\{ \hat{\boldsymbol{z}}_j^0, \ldots, \hat{\boldsymbol{z}}_j^i, \ldots, \hat{\boldsymbol{z}}_j^{N_j} \right\}_{j=0}^{n_{\mathcal{Z}}}$ with samples $\hat{\boldsymbol{z}}_j^i$ taken at times $i\,\Delta t$
- max lag $N_{max}$
- $p_{\text{cond}}$ probability of conditional training

2: Initialize UPT++ flow matching model $v_\theta$

3: $\mathcal{Z}' = \text{Concatenate}\left(\left\{ \hat{\boldsymbol{z}}_j^0, \ldots, \hat{\boldsymbol{z}}_j^{N_j - N_{\max}} \right\}_{j=0}^{n_{\mathcal{Z}}}\right)$

4: **while** not converged **do**

5: $\qquad \hat{\boldsymbol{z}}_j^i \sim \text{Choice}\left(\mathcal{Z}'\right)$

6: $\qquad N \sim \text{DiscreteUniform}(1, N_{\max})$

7: $\qquad (\tilde{\boldsymbol{z}}_{\text{cond}}, \tilde{N}_{\text{cond}}) \leftarrow (\hat{\boldsymbol{z}}_j^i, N)$ with probability $p_{\text{cond}}$ else $\emptyset$

8: $\qquad \tilde{s} \sim \text{ContinuousUniform}(0, 1)$

9: $\qquad \tilde{\boldsymbol{z}}_0 \sim \mathcal{N}(0, 1)$

10: $\qquad \tilde{\boldsymbol{z}}_s \leftarrow (1 - \tilde{s})\tilde{\boldsymbol{z}}_0 + \tilde{s}\hat{\boldsymbol{z}}_j^{i+N}$

11: $\qquad$ Take gradient step on $\nabla_\theta \left\| v_\theta(\tilde{s}, \tilde{\boldsymbol{z}}_s, \tilde{\boldsymbol{z}}_{\text{cond}}, \tilde{N}_{\text{cond}}) - \left(\hat{\boldsymbol{z}}_j^{i+N} - \tilde{\boldsymbol{z}}_0\right) \right\|^2$

12: **end while**

13: **Output:**
- UPT++ flow matching model $v_\theta$

---

**Algorithm 2** Sampling UPT++ sampling operator

1: **Inputs:**
- trained UPT++ flow matching model $v_\theta$
- Condition state $z_{\text{cond}}$
- Forward sampling timesteps $N_{\text{cond}}$
- Number of ODE steps $N_{\text{ODE}}$
- Guidance parameter $\omega$

2: $\tilde{z}_0 \sim \mathcal{N}(0, 1)$
3: $h \leftarrow \frac{1}{N_{\text{ODE}}}$
4: $v_{\theta,\text{guided}}(.,.) \leftarrow (1 - \omega)\, v_\theta(.,.,\emptyset) + \omega\, v_\theta(.,.,z_{\text{cond}}, N_{\text{cond}})$
5: **for** s=1,…,$N_{\text{ODE}}$ **do**
6:      $\tilde{z}_{s\,h} \leftarrow \text{ODEStep}(v_{\theta,\text{guided}}((s-1)\,h, \tilde{z}_{(s-1)\,h}), h)$
7: **end for**
8: **Output:**
- Sample $\tilde{z}_1$

### B.4 Molecular Graph Reconstruction

We present a systematic procedure for reconstructing the molecular graph, from the predicted density distribution. Figure B1 provides a visual representation of certain steps.

1. Evaluate positions on our occupancy field and remove all values below a threshold of 0.5 (used throughout our experiments). Note that reconstructing the molecule based solely on the density channels yields similar results.

2. Peak finding: Identify local maxima in the thresholded occupancy map using a maximum filter.

3. For each density channel, select the top $N_\alpha$ values, where $N_\alpha$ is the expected number of atoms of element $\alpha$ in the molecule.

4. Reconstruct bonds [2] using OpenBabel (O'Boyle et al., 2011).

5. Validate the chemical equivalence of the encoded molecule using InChI codes (Landrum, 2016).

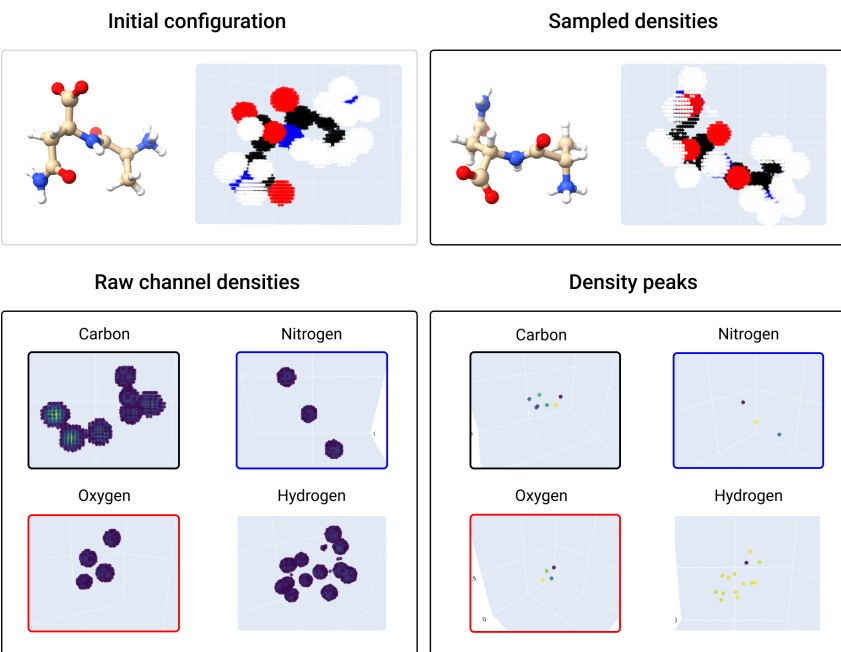

Figure B1: **Top row**: Densities across all channels in a single plot, distinct colors represent different atomic channels. All values exceeding an occupancy threshold are assigned a uniform value. **Top left**: Original molecular conformation obtained by applying the encoder/decoder only. **Bottom left:** Raw density maps for each channel, predicted by our latent sampling operator. The density values increase concentrically towards the atomic positions. **Bottom right**: Extracted peaks indicating atomic positions.

---

[2]https://github.com/guanjq/targetdiff/blob/main/utils/reconstruct.py

## B.5   ADDITIONAL RESULTS

In Figures B2, B3, we visualize results corresponding to those in Table 2 analogously to Figure 6.

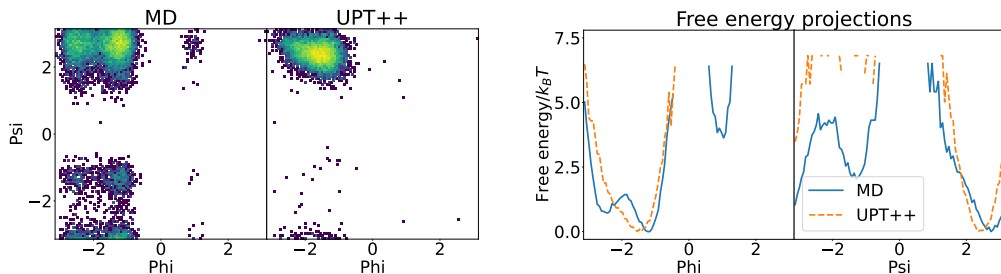

Figure B2: **2AA: AN (3.5).** *Left half:* Ramachandran plots comparing 10k UPT++ and 9.8k MD samples. *Right half:* Free energy surface of the same UPT++ and MD samples.

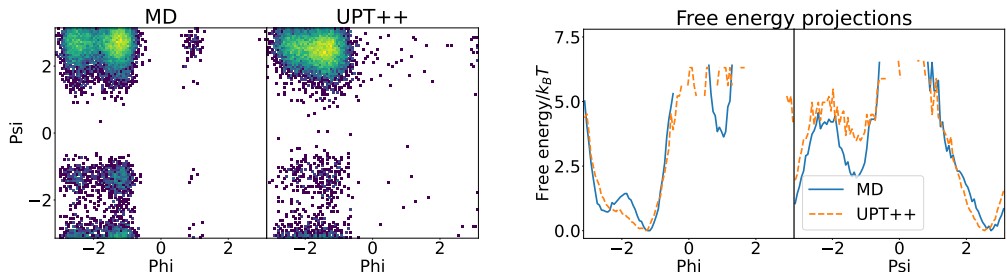

Figure B3: **2AA: AN (1.0).** *Left half:* Ramachandran plots comparing 10k UPT++ and 9.8k MD samples. *Right half:* Free energy surface of the same UPT++ and MD samples.

