# OpenReview forum: "UPT++: Latent Point Set Neural Operators for Modeling System State Transitions"
_ICLR.cc/2025/Conference — ICLR 2025 Conference Withdrawn Submission_

### Official Review · Reviewer_Aaq8 · 2024-10-23

**Soundness:** 3
**Presentation:** 2
**Contribution:** 2
**Rating:** 5
**Confidence:** 3

**Summary:**

The paper proposes a particle-based neural PDE model, where fields are represented by importance-sampled pointclouds, and autoencoded into latent space. The key contribution is improving runtime speed of neural PDEs.

**Strengths:**

- The writing has a good flow, and gives a good overview of the field.
- The results show dramatic speedups: the method seems to work well.
- The paper expands the method from fluid simulations to atom samplings, which is a nice extension.

**Weaknesses:**

W

- As far as I understand, the contributions are modest. The main contribution seems to be sampling from the fields according to their density (instead of uniform). This is clearly a useful contribution, but an obvious one, and it’s difficult to believe that this hasn’t been done before. The rest of the method seems taken from from literature.
- The clarity has issues. The method presentation is insufficient with parts of the model being just textually described with no math, and parts of the model have math but with insufficient exactness. The paper doesn’t do a good job at describing how the proposed method positions in the domain wrt other methods. The experiments are black-box.
- The experiments are insufficient. The main contribution of this method is scaling to larger particle systems. The paper lists three earlier approaches to scale GNN models, but only compares to one. The paper needs to compare to all prominent earlier scaling-up methods in this domain; and to the UPT; and to some neural operator type methods or neural PDE solvers.
- The molecular sampling experiments do not have any baselines, and the results seems a bit disappointing.
- The method speedup is only demonstrated in static experiments. The paper should show how the methods scale to different particle counts, and show the benefits of operating with more/less particles in speed or accuracy. The significance of the speedup needs some arguments (why do we care?).
- The results lack ablations, and little insight is drawn into what the system has learnt, or why, or what part is doing what. Some key things to explore could be the sampling size, and latent space size.

**Questions:**

Q

- The notation blurp at 207 is a bit confusing. Is $u$ a function object, or an evaluation of a function at one point, or is this a function evaluated at multiple points (the boldfacing hints towards this)? What is $U$? Is this a space of functions, where each element is one function; or is this a single function where each element is one evaluation of this function at one point?
- I’m now assuming that $u$ is a function, ie. I would write it as $u(\cdot)$. The encoder needs to take in a function, and output a matrix. Can you elaborate how is this possible in theory or practise? How can you input a function into your encoder, or how can you output a function from decoder? I’m suspecting that there is a trick here that is not made explicit, such as outputting a symbolic function, or perhaps a kernel function coefficients, or something like that. However, the notation to me clearly disallows replacing the functions with their parameterisations, which is even more baffling. The 237 is talking about decoding pointwise, but this is exactly what shouldn’t happen: the decoder outputs a function, not function evaluations. Later in 250 the encoder becomes some kind of matrix encoder. Err… Can you help me understand? The math does not seem precise.
- I don’t understand what is u_k^t. What is k? Are these points, or sets of points? What is a “node”? What is “selected”? How do you select? Why? The importance sampling was barely described at all. This needs proper math, and/or an algorithm box.
- “Next, via *local aggregation* we propagate neighboring information to the respective supernodes (radius graph for connectivity to keep discretization convergence), and finally *global information aggregation* pools the information into a fixed size and uniform latent space via perceiver blocks (Jaegle et al., 2021a;b).” I can’t follow what this stuff means. Are you feeding some kind of point cloud through a GNN? Wasn’t the input defined as a function (not point cloud)? What is “fixed size and uniform latent space”? How can a space be of fixed size? What does this even mean? Surely a space is always fixed size…? What is uniform space? What are supernodes? What are nodes?
- I think you redefined u^t somewhere, since in 257 the u^t is likely some kind of point matrix and not a function anymore.
- I dont’ really understand how the encoder works. The pointcloud can be bigger than the latent representation, so you need to somehow output a smaller code. Having an algorithm box would help.
- “First, via *occupancy decoding* (Mescheder et al., 2019), we decode an occupancy field to identify where particles or atoms are located in space. Secondly, we *point-wise decode* a field quantity and consider only the occupied points” I don’t understand these sentences. How do you decode an occupancy field? How do you represent the field? Where does the occupancy field go or what does it do? Is it part of $u$? All the math is missing here.
- How do you only “consider only the occupied points” What does this even mean? How do you “consider”? There are no “points” in the latent space, so where do they come from?
- The text hints that there are some output positions and queries and keys and values. Err… no math is given about these, so I’m pretty lost. I think this part is exclusive for people who have read Alkin 2024, but I’m not sure if it’s a good idea to limit your audience to them.
- What are “positive and negative learning signals”?
- How do you “sample points at all locations of the domain”? Surely this is infinitely many points…? There’s no math so I’m unsure what’s going on.
- The peak finding and openbabel stuff is not understandable from the presentation.
- How can A operate R^k if the latent space is R^(nxk)? Is the A batched? Does it make sense to ignore the tokens? In 290 the paper redefines z as h-dimensional vector, while previously in 213 it was a matrix. Err….. what? I’m a bit unsure at this point if $z$ represents points or functions. Oh wow, the next sentence in 295 flips the z back to matrix.
- I could not follow the 306 paragraph. It seems that the successor sampling is done with normalizing flows, which is a bit strange. Why normalizing flows? It seems that the latent system is deterministic, so why do you call this “sampling”? A CNF would just forward the z^t under some ODE until it hits z^t’. There is no sampling here. I also don’t understand how you learn this, or against what data. This entire part feels loose, and I’m not convinced that any kind of proper conditional density sampling is actually happening.
- Wha does “inference” mean?
- I don’t understand the training setting. What is known, what is data? Where do you sample? For instance in the Navier-stokes: which timepoint are you sampling from? Where do you get the function you are sampling? Which locations are you sampling? Do you know the system? What are you trying to achieve or what is the goal? What are unknowns? Are you solving the initial value problem, or the system discovery problem, or something else? I’m confused what is the task you are solving in this paper.
- What does sampling different input and output points mean? Are the “output” points the values $u$, or just different input points? I’m so confused. Please please use math to define what’s going on. Can you define what do you mean by “point”? Is this x?
- The training is not sufficiently described. Eg. the losses are not even in the appendix (!).
- I’m confused what is the novelties of the method. The training seems pretty standard, the approximator seems pretty standard, and the encoder/decoder seem just GNNs which are common. Maybe the occupancy/velocity parameterisation could be novel, but this is quite system-specific and a simple idea. I assume the density-sampling is novel in this domain, although a common idea.
- In experiments there is still some method development going on, with a remark about k=4. Please define your model in one place. I’m not sure what the “two pointclouds” are, or why are we doing some nearest neighbors here.
- It’s not very clear how your method differs from the GNS baseline. Having a comparison table would help. I think the main innovation is that UPT++ just samples less points, and otherwise follows the GNS/GNN approach. I’m probably wrong here.
- Results needs standard deviations
- The conclusion again emphasises the fixed-size aspect. Can you elaborate? Why is this important? Isn’t every autoencoder model fixed-size?
- The conclusion says that UPT is not mass-preserving, while GNNs are. I thought that UPT is a GNN method. Can you elaborate?

---

### Official Review · Reviewer_Sgwg · 2024-10-28

**Soundness:** 2
**Presentation:** 2
**Contribution:** 2
**Rating:** 3
**Confidence:** 3

**Summary:**

In this paper, the authors present UPT++, a framework for modeling the dynamical evolution of physical systems which obey mathematical models, including the Navier-Stokes and Schrodinger equations. The core contention is that while particle-based discretization schemes are traditionally used to model these phenomena, deep learning techniques are better suited to learn continuous representations. UPT++ addresses this disparity by encoding physical systems into a continuous latent space, in which a time evolution or sampling operator is learned. A decoder is then used to map back to the data space. The method is demonstrated on fluid and molecular dynamics tasks, and is demonstrated to be faster, more scalable, and more accurate relative to existing methods.

**Strengths:**

The idea of learning dynamics in the latent space, while not new, is a promising and interesting direction particularly as ML methods scale to more complicated dynamical systems. Regarding presentation, the paper is fairly well-written, and the figures do a strong job of conveying the core of the method.

**Weaknesses:**

As I am more familiar with the ML for molecular dynamics field than ML for PDEs/fluids, I will focus more on that aspect of the work in my review. In short, I found the MD section to be quite underdeveloped, in terms of methodology, exposition, and empirical results. I will elaborate below.

1. As I understand it, the primary motivations for performing dynamical propagation in a latent continuous space are that it might be a more suitable inductive bias for learning smooth/continuous dynamics (which could then enable taking larger timesteps, etc.), and that it can improve memory/compute efficiency due to the latent bottleneck. However, in the MD experiments, none of these points are studied in any depth. The authors present results on alanine dipeptide and small peptides of up to 50 atoms. These systems do not pose scaling challenges due to the low number of atoms, and as far as I could tell, there was no mention of runtime in the MD experiments. In addition, while alanine dipeptide is known to have transition barriers which make the conformational space somewhat challenging to navigate, I do not know if the same can be said for the other proteins. There was also no discussion of varying the timestep of simulations (e.g. does mapping to a continuous latent space admit larger timesteps due to the smoother dynamics?). In short, there is no discussion of what the value-add of continuous latent space modeling is in this context; the experiment is at-best a proof-of-concept that the proposed method can work, but not that it is useful or better than existing methods. Some experiments that could be done to strengthen this section may include: quantifying whether the dynamics are somehow "smoother" or more well-behaved in the latent space than the data space, as well as scaling to realistic systems (e.g. those considered in the Allegro work) in which speed/memory efficiency are big challenges.

2. The authors mention in passing that a MD/energy minimization step is required after each step of forward propagation to mitigate the distribution shift that comes with autoregressive modeling. To me, this is a serious limitation that undermines the usefulness of the approach. I would like to see much more details/elaboration on this point: how many energy minimization steps are performed at every sampling step? How much does it contribute to the runtime? For how many steps can the model simulate stably without this modification?

3. There are no baselines for comparison in the MD section. Some options that immediately come to mind include MLCGMD [1] and machine learning force fields (MLFFs)  like MACE [2] and NequIP [3]. Without comparison to these baselines, it is impossible contextualize important metrics like long-time simulation stability, reproduction of observables, inference speed, etc.

Some general comments applicable to both MD and fluids:

1. I would have liked to see some analysis/ablations on the effect of the latent bottleneck on performance. How much can the latent space be compressed and still achieve stable/accurate dynamics? What is the uncertainty introduced by mapping to a latent space, and how does this manifest in the final predictions?

2. One of the central motivations for the proposed method is that deep learning excels at learning continuous representations. However, there are numerous examples in other areas of ML where discretization has proven to be very useful for generative modeling of continuous data modalities. For example, it is fairly common to discretize images via vector-quantization [4] and perform autoregressive modeling over discrete tokens. There are examples of this in protein modeling [5] and reinforcement learning [6], among many others. Can the authors comment on these approaches and how their "discretization paradox" fits into this framework?

[1] Fu, Xiang, et al. "Simulate time-integrated coarse-grained molecular dynamics with multi-scale graph networks." Transactions on Machine Learning Research (2023).

[2] Batatia, Ilyes, et al. "MACE: Higher order equivariant message passing neural networks for fast and accurate force fields." Advances in Neural Information Processing Systems 35 (2022): 11423-11436.

[3] Batzner, Simon, et al. "E (3)-equivariant graph neural networks for data-efficient and accurate interatomic potentials." Nature communications 13.1 (2022): 2453.

[4] Razavi, Ali, Aaron Van den Oord, and Oriol Vinyals. "Generating diverse high-fidelity images with vq-vae-2." Advances in neural information processing systems 32 (2019).

[5] Hayes, Tomas, et al. "Simulating 500 million years of evolution with a language model." bioRxiv (2024): 2024-07.

[6] Shridhar, Mohit, Lucas Manuelli, and Dieter Fox. "Perceiver-actor: A multi-task transformer for robotic manipulation." Conference on Robot Learning. PMLR, 2023.

**Questions:**

Boltzmann generators are mentioned several times in the paper, but these differ from the proposed approach in that they produce i.i.d samples from the Boltzmann distribution rather than dynamical evolution trajectories. Can the authors' proposed approach be repurposed for i.i.d sampling as well?

---

### Official Review · Reviewer_bE58 · 2024-11-03

**Soundness:** 3
**Presentation:** 2
**Contribution:** 2
**Rating:** 3
**Confidence:** 4

**Summary:**

Authors present UPT++, which is an extension of the Universal Physics Transformer (UTP). This extension focuses on improving the efficiency of UPT by introducing importance-based decoding and encoding, and more efficient sampling via flow matching.

**Strengths:**

- Importance-based Encoder has two more layers of compression compared to the original UPT paper. UPT only includes compression via local aggregation compared to UPT++’ importance-based sampling and perceiver-based latent encoding.

**Weaknesses:**

- Limited comparisons: The authors do not use other neural operator models to see if they match their sampling efficiency. For example Neural Integral Equations, FNO, U-Net and Neural Operators. In addition, other more traditional datasets for PDEs would be useful.
- Efficient ablations: It is not clear what component of UPT++ is the one that gives it its improved MSE and speed. Ablation studies would be a great addition.
- Given the focus on different sampling resolutions, it would have been great to see the ability of the model to interpolate in space. Meaning train the model at one spatial resolution and generate data at a higher one. This has been shown in other models to be a learnable ability of Neural Operator models.
- Writing requires improvement, particularly the introduction section, where there seems to be a disconnect between these paragraphs, as the ideas don't appear to lead naturally from one to the other.

**Questions:**

Suggestions:

- Improve writing on the introduction. Given comments above.
- Introduce models such as FNO, NIE, and U-Net into your comparisons.
- Ablation studies to compare your improvement over UPT.

---

### Official Review · Reviewer_Fc6b · 2024-11-11

**Soundness:** 2
**Presentation:** 1
**Contribution:** 2
**Rating:** 3
**Confidence:** 3

**Summary:**

The paper proposes a framework to represent dynamical systems involving interacting particles as continuous systems, to facilitate their integration using tools from ML. This encoding from discrete to continuous (and the associated decoding from continuous to discrete) is done using the framework of neural operators, and in particular via the Universal Physics Transformers introduced recently. Some numerical examples involving fluid flows and molecular dynamics applications are given to illustrate the methodology.

**Strengths:**

The general aim of the paper is ambitious and would be useful if achievable.

**Weaknesses:**

The main issue with the paper is that it is quite vague and at the end gives few concrete details about the way the ambitious objectives listed are achieved. More specifically:

The paper starts with an introduction about PDEs and their usage in Physics, which at the end is quite generic and a bit pompous (is it really necessary to mention the Millennium problems here?). The authors also make the point that both the Navier Stokes equation and the Schroedinger equation (their two primary example in that section) needs discretization to be solved, thereby justifying what  they call the `discretization paradox': however, particle methods of the type they discuss are only a very small subset of the numerical schemes introduced to solve such equations.

The section BACKGROUND: PARTICLE METHODS AND NEURAL OPERATORS is wordy, and at the end rather short on concrete details. The same can also be said about the section UPT++ , which presents a framework that is rather new and not so well-established, and should therefore be explained in more details. It is rather hard to figure on what si actually done in practice after reading Secs. 3.1, 3.2, and especially 3.3. Sec. 3.4 is also quite short and vague considering that the authors claim that their method is a universal methodology ...


Section 4 is supposed to present the main examples but here too it is hard to decipher what is actually done, and how the approach compares in terms of efficiency and accuracy with existing methods.

**Questions:**

1. Can the authors explain in more detail what UPT++ is and what are the advantage it confers in the present setup? It would be good to do this more concretely on one of the numerical examples chosen for illustration.

2. Can the importance-based decoding and encoding be also explained in more details? It seems that the approach is based on representing the particles with density fields, but it is not obvious how to do so e.g. in molecular systems involving peptide since the atoms in these systems are not interchangeable and so their individual identities matters. Can the authors explain how they bypass this difficulty and keep the necessary knowledge of the atomistic details in their approach?

3. It seems that the approach learns an evolution operator in the latent space of the continuous representation. How is this done in practice, i.e. using which loss? How does the procedure guarantees fidelity with respect to the known evolution laws at particle level? Is the procedure stable over roll-outs, as it should be? What about its accuracy?

4. How is the sampling step using flow matching done in practice? Does this involve learning another velocity field, as is done with flow matching/stochastic interpolants/rectified flows? Does the approach proposed simply build on these methods, or does it add to them? If so in which way?

5. Can the author please specify that are the training losses they use?

6. What exactly are the numerical methods against which the example in Sec. 4.1 is benchmarked against. Simulating fluids with moving boundaries has a huge history, and can be done using the finite elements method, the immersed boundary method, etc. A more thorough comparison with these existing methods is needed.

7. The example in Sec. 4.2 seem to be benchmarked against standard MD simulations. Since the latter are rather straightforward, and can be combined with are events sapling methods to reach longer time scale on which conformation changes occur, it would be useful to have a more thorough discussion of the advantages conferred by the method proposed by the authors. Some implementations details would also be welcome. The results shown in Fig 6 also show that the approach proposed seems rather inaccurate, with errors in the free energy of several kT, which is worrisome considering how toyish  the AD example studied is.

---

### Note · Authors · 2024-11-15

**Comment:**

We thank all reviewers for their feedback and for their constructive comments and suggestions.
We have decided to withdraw our submission. We plan to address the reviewers’ feedback thoroughly and work towards an improved version for a future submission.

**Withdrawal Confirmation:**

I have read and agree with the venue's withdrawal policy on behalf of myself and my co-authors.